# Actin cytoskeleton and complex cell architecture in an Asgard archaeon

Thiago Rodrigues-Oliveira[1,4], Florian Wollweber[2,4], Rafael I. Ponce-Toledo[1,4], Jingwei Xu[2], Simon K.-M. R. Rittmann[1], Andreas Klingl[3], Martin Pilhofer[2✉] & Christa Schleper[1✉]

Asgard archaea are considered to be the closest known relatives of eukaryotes. Their genomes contain hundreds of eukaryotic signature proteins (ESPs), which inspired hypotheses on the evolution of the eukaryotic cell[1–3]. A role of ESPs in the formation of an elaborate cytoskeleton and complex cellular structures has been postulated[4–6], but never visualized. Here we describe a highly enriched culture of 'Candidatus Lokiarchaeum ossiferum', a member of the Asgard phylum, which thrives anaerobically at 20 °C on organic carbon sources. It divides every 7–14 days, reaches cell densities of up to $5 \times 10^7$ cells per ml and has a significantly larger genome compared with the single previously cultivated Asgard strain[7]. ESPs represent 5% of its protein-coding genes, including four actin homologues. We imaged the enrichment culture using cryo-electron tomography, identifying 'Ca. L. ossiferum' cells on the basis of characteristic expansion segments of their ribosomes. Cells exhibited coccoid cell bodies and a network of branched protrusions with frequent constrictions. The cell envelope consists of a single membrane and complex surface structures. A long-range cytoskeleton extends throughout the cell bodies, protrusions and constrictions. The twisted double-stranded architecture of the filaments is consistent with F-actin. Immunostaining indicates that the filaments comprise Lokiactin—one of the most highly conserved ESPs in Asgard archaea. We propose that a complex actin-based cytoskeleton predated the emergence of the first eukaryotes and was a crucial feature in the evolution of the Asgard phylum by scaffolding elaborate cellular structures.

Soon after the discovery of archaea as a separate lineage besides bacteria, molecular and phylogenetic studies suggested that there is a deep common evolutionary descent between archaea and eukaryotes[8,9]. However, only recently has the discovery of the first Lokiarchaeota[10] (now Lokiarchaeia[11]) and the wider superphylum of Asgardarchaeota in metagenomic analyses[1,2,11–17] corroborated a distinct relationship and a possible direct emergence of eukaryotic cells from archaea. In fact, eukaryotes form a direct sister group to Asgardarchaeota or even arise within Asgardarchaeota in most phylogenomic analyses[3,10,18].

Compellingly, all members of the Asgardarchaea carry an extensive repertoire of genes that were originally assumed to be unique to eukaryotes (ESPs)[1–3,10,19,20]. These ESPs are mostly associated with features of cells with high complexity, such as cytoskeleton formation, transport and the shaping of membranes. For example, the observation that Asgard archaeal genomes encode a complete and functional ubiquitin-coupled ESCRT system[10,21] suggested the possibility of elaborate intracellular membrane compartments[10]. Another notable example is genes encoding several close homologues of eukaryotic actin. While F-actin-like assemblies have been identified in other archaea[22,23], Asgard archaea also possess actin-related proteins (ARPs), as well as actin-binding proteins. Notably, Asgard profilins and gelsolins were found to be able to modulate the dynamics of eukaryotic actin[5,19,24–26],

indicating the existence of an elaborate and dynamic cytoskeleton[4]. However, the in situ structures and functions of archaeal actins remain unclear.

A seminal study[7] presented the first enrichment culture of an Asgard archaeon, 'Candidatus Prometheoarchaeum syntrophicum', which grows slowly to low cell densities in syntrophic consortia with molecular-hydrogen-consuming organisms. As 'Ca. P. syntrophicum' cells show long branched protrusions, the authors proposed a hypothesis for eukaryogenesis, in which a primordial Asgard archaeon closely interacts with the predecessor of the bacterial endosymbiont and eventually endogenizes it[7,27]. These observations were consistent with the stepwise mechanism of eukaryogenesis that was first proposed as the 'inside out' model[28]. The role of Asgard ESPs could so far not be investigated in the natural host, making it difficult to further test these conceptual models. Although 'Ca. P. syntrophicum' was shown to transcribe ESPs, characteristics regarding its intracellular architecture could not be revealed. Fundamental questions regarding the presence of a cytoskeleton or internal compartmentalization in Asgard archaea remain unclear, as does the structure of the cell envelope. Here we combine the enrichment of an experimentally tractable Asgard archaeon with state-of-the-art imaging to reveal its cellular architecture at macromolecular detail.

[1]Department of Functional and Evolutionary Ecology, Archaea Biology and Ecogenomics Unit, University of Vienna, Vienna, Austria. [2]Institute of Molecular Biology & Biophysics, ETH Zürich, Zürich, Switzerland. [3]Plant Development & Electron Microscopy, Biocenter, Ludwig-Maximilans-Universität München, Planegg-Martinsried, Germany. [4]These authors contributed equally: Thiago Rodrigues-Oliveira, Florian Wollweber, Rafael I. Ponce-Toledo. ✉e-mail: pilhofer@biol.ethz.ch; christa.schleper@univie.ac.at

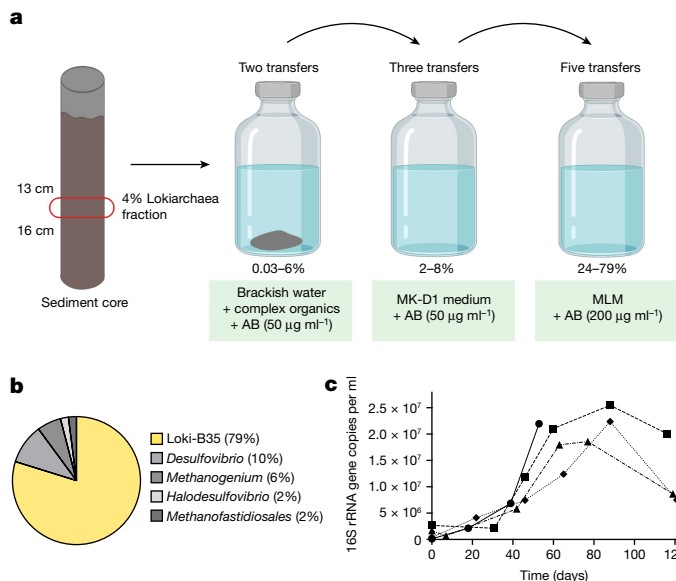

**Fig. 1 | Enrichment and cultures of Loki-B35. a**, Schematic of our cultivation approach. A sediment core fraction was used as an inoculum for cultivation in sterile-filtered environmental water from the sampling site supplemented with complex organic compounds. The enrichment was then transferred to modified MK-D1 medium[7]. Enrichments of up to 79% were obtained when the cultures were transferred to MLM supplemented with casein hydrolysate. AB, antibiotics. The figure was created using BioRender. **b**, The composition of the culture with the highest enrichment as assessed by 16S rRNA gene amplicon sequencing. **c**, Growth curves ($n = 4$) of Loki-B35 in MLM (80:20 $N_2$:$CO_2$) supplemented with casein hydrolysate (growth was quantified by qPCR), indicating maximum cell densities of $2.5 \times 10^7$ per ml and generation times of about 7–14 days.

## A Lokiarchaea culture from sediment

Considering that lokiarchaeal organisms and other Asgard archaea can be found in a variety of anoxic and often marine environments[29,30], we screened DNA from shallow-water sediment from different locations for the presence of 16S rRNA genes of Asgard archaea to select suitable and easily reachable sampling sites for establishing enrichments. Sediments from a small estuarine canal that regularly receives water from the Mediterranean near the coast of Piran, Slovenia, were identified to have the highest relative abundance at the 13–16 cm depth layer, exhibiting up to 4% of Asgard archaea 16S rRNA genes in amplicon sequencing (Extended Data Figs. 1 and 2).

The identified sample was used to inoculate enrichment cultures (Fig. 1a) with media of different compositions and various headspace conditions (Supplementary Table 1). With periodic monitoring using quantitative PCR (qPCR) with Lokiarchaea-specific primers (Extended Data Fig. 3), growth could be observed after 140 days at 20 °C in serum flasks containing sterile-filtered water from the original source supplemented with complex organics (casein hydrolysate, milk powder and amino acids). However, after two transfers under these conditions, growth could no longer be detected and a second round of screening with different medium compositions was performed. Using a modification of the medium MK-D1 reported for the cultivation of '*Ca.* P. syntrophicum'[7], cell growth recovered, and abundances reached repeatedly 2–8%. However, higher enrichments, were not achieved under these conditions. Only through developing a minimal medium, mostly by reducing the input of organic carbon sources to a single compound and by increasing antibiotic concentrations, lokiarchaeal relative abundances reached between 25% and 80% after several transfers. The highest enrichments were achieved in minimized lokiarchaeal

medium (MLM) with casein hydrolysate (Fig. 1a), while growth was also observed with either tryptone, peptone, milk powder, single amino acids, glucose or pyruvate (Supplementary Table 1).

Amplicon sequencing analyses of 16S rRNA genes revealed that the culture with the highest enrichment (Loki-B35) consisted of three dominant and two minor species: a single Lokiarchaeon sequence (79%), a sulfate-reducing bacterium of the *Desulfovibrio* genus (10%), a hydrogenotrophic methanogen of the *Methanogenium* lineage (6%), as well as a *Halodesulfovibrio* and a member of the *Methanofastidiosales* genus (both at around 2%) (Fig. 1b and Extended Data Fig. 1). Notably, both *Halodesulfovibrio* and *Methanogenium* were also syntrophic partners in the enrichment cultures of '*Ca.* P. syntrophicum'[7], which stems from a geographically different and deep-sea environment. Thus, it seems probable that both Lokiarchaea rely on a similar metabolism, involving the fermentation of peptides to $H_2$ and/or small organic acids.

Loki-B35 grows without lag phase to maximum cell densities of up to $2.5 \times 10^7$ cells per ml, sometimes even $5 \times 10^7$ cells per ml within 50 to 60 days (Fig. 1c) when started with a 10% inoculum. However, extremely long lag phases of 90 to 120 days were observed when the inoculum originated from stationary cultures. The generation time was estimated to be 7 to 14 days. Compared with the deep sea Lokiarchaeon '*Ca.* P. syntrophicum' (maximum cell densities of $10^5$ cells per ml, generation time of 14–25 days)[7], Loki-B35 grows considerably faster and to higher cell densities.

## Loki-B35 genome analyses and phylogeny

The genome of Loki-B35 was assembled into one contig based on short and long read sequencing. It contains 6,035,313 base pairs (bp), encoding 5,119 predicted proteins, three 16S and 23S ribosomal rRNA copies (two of them in operons) and 34 tRNAs (Fig. 2a). The presence of only one rRNA operon in the closed genomes of '*Ca.* P. syntrophicum' and two Heimdallarchaea[7,31], but two operons in the high-quality assembly of the Lokiarchaeota genome '*Candidatus* Harpocratesius repetitus' FW102 (ref. [31]) and even three ribosomal RNA operons in Loki-B35, indicates that there is variability that may be based on the strains' generation time or flexibility to adapt to changing environmental conditions[32,33].

Compared with '*Ca.* P. syntrophicum', the genome of Loki-B35 is significantly larger (by approximately 1.6 Mb), which is also reflected by 2,256 unique proteins (Fig. 2b). Functional annotation of orthogroups revealed that genes of Loki-B35 are enriched in almost all categories, reflecting its overall larger genome size (Extended Data Fig. 4 and Supplementary table 2). Notably, the fraction of proteins representing ESPs scales approximately with genome size, representing 5.5% in '*Ca.* P. syntrophicum' (218 ESPs) and 5% in Loki-B35 (258 ESPs; calculated according to the most recent asCOG database[2]) (Supplementary table 3). Compared to '*Ca.* P. syntrophicum', the genome of Loki-B35 is particularly enriched for genes associated with membrane trafficking and protein transport (Fig. 2c).

The sequence similarity of the 16S rRNA genes of Loki-B35 and '*Ca.* P. syntrophicum' is 95.3% and the common orthologous proteins share 58.4% amino acid identity, which justifies a separation into different genera[34,35]. This separation is supported by the large number of unique proteins that represent 47.7% of the complete predicted protein set. The evolutionary distance between the two cultivated Lokiarchaea also becomes evident in a gene synteny comparison, which shows a high number of rearranged genes and genome-specific regions being distributed throughout the genomes (Extended Data Fig. 5).

In a universal phylogeny based on 23 conserved ribosomal proteins from 291 representative species of all three domains of life, all Asgard archaea formed a monophyletic group with eukaryotes as their direct sister lineage (Fig. 2d). The Asgard phylogeny (Fig. 2e), based on 168 genomes, clearly separates all described classes of the phylum and is consistent with other recent phylogenomic analyses[1,2,13], although

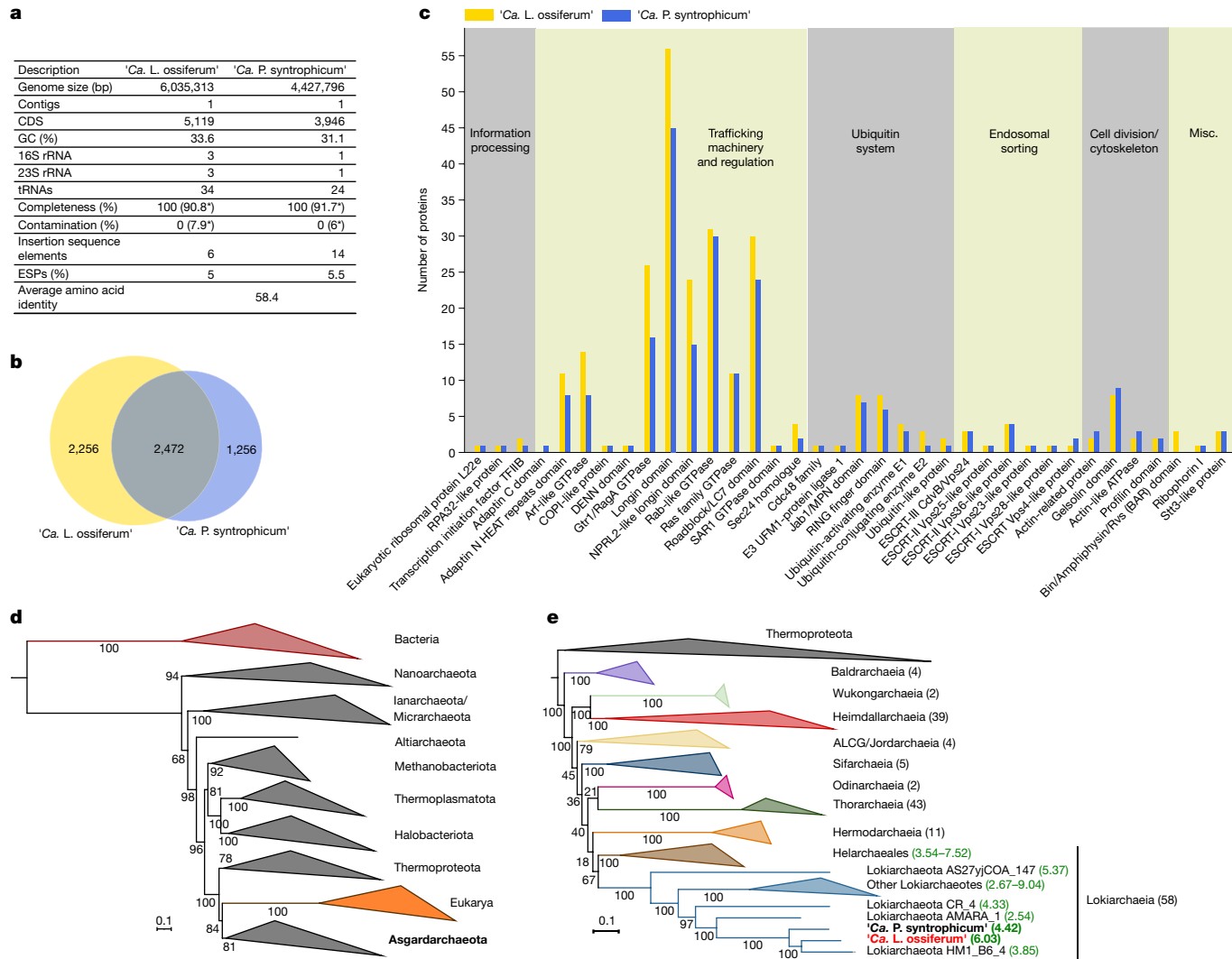

**Fig. 2 | Genome analysis and phylogenetic placement of '*Ca.* L. ossiferum'.**
On the basis of the analysis of the closed genome of enrichment Loki-B35, we propose a description of the species '*Ca.* L. ossiferum'. **a**, The characteristics of the genome of '*Ca.* L. ossiferum' in comparison to '*Ca.* P. syntrophicum'. Note the substantial difference in genome size. The values indicated by asterisks are the estimated values of contamination and completeness on the basis of the identification of marker genes performed by CheckM[53] (Methods). **b**, The diagram shows to scale the number of shared clusters of orthologous proteins between '*Ca.* L. ossiferum' and '*Ca.* P. syntrophicum' as well as genome-specific clusters. A more detailed analysis is provided in Extended Data Fig. 4. **c**, A comparison of the occurrence of ESPs in '*Ca.* L. ossiferum' and '*Ca.* P. syntrophicum'. Annotation of ESPs was performed according to the asCOG database[2]; general functional categories (on top) were added similar to earlier assignments[1]. Note that '*Ca.* L. ossiferum' is enriched for ESPs of the following protein families associated with trafficking machineries: adaptin N heat repeats domain; Arf-like GTPase; Gtr1/RagA GTPase; longin domain; NPRL 2-like longin domain. **d,e**, Maximum-likelihood (ML) phylogenies based on the concatenation of 23 universally conserved ribosomal proteins[2]. **d**, Tree of life showing Eukarya as a sister clade of Asgardarchaeota. **e**, '*Ca.* L. ossiferum' and '*Ca.* P. syntrophicum' belong to the same Lokiarchaeia class. Taxonomic assignments are based on the recently proposed classification[11]. Branch supports were calculated with 1,000 ultrafast bootstrap samples. The values in square brackets show the genome sizes of complete genomes (bold) and MAGs within Lokiarchaeia.

inner branching nodes vary depending on differences in the analyses and datasets. Loki-B35 together with '*Ca.* P. syntrophicum' and three other lokiarchaeal metagenome-assembled genomes (MAGs) formed one of two major sublineages within the class Lokiarchaeia (Fig. 2e). On the basis of these analyses, we propose a new genus and species for which we propose the name: '*Ca.* L. ossiferum' strain Loki-B35 (see below for etymology).

## Identification of '*Ca.* L. ossiferum' cells

Our next goal was to identify individual '*Ca.* L. ossiferum' cells and characterize them using microscopy. Fluorescence in situ hybridization (FISH) with specific probes showed that '*Ca.* L. ossiferum' cells appeared as spheres of variable cell size (0.3–1.0 μm), being considerably smaller than other organisms in the enrichment (Fig. 3a). Cell counts from FISH analysis (not shown) confirmed the relative abundance of up to almost 80% in our highest enrichments as seen by amplicon sequencing (Fig. 1b).

We next plunge-froze cells of a live culture onto electron microscopy (EM) grids to image them in a near-native state using cryo-electron tomography (cryo-ET). A major challenge was the low cell density combined with the high fragility of the '*Ca.* L. ossiferum' cells, which did not allow us to perform any processing steps. Thus, instead of concentrating the samples, we performed an extensive screening of the grid by recording two-dimensional (2D) overview images, followed by cryo-ET data collection of selected cells. This approach revealed three general cell types that had distinct morphologies and cell envelope architectures (Fig. 3b,c). One class consisted of round-shaped

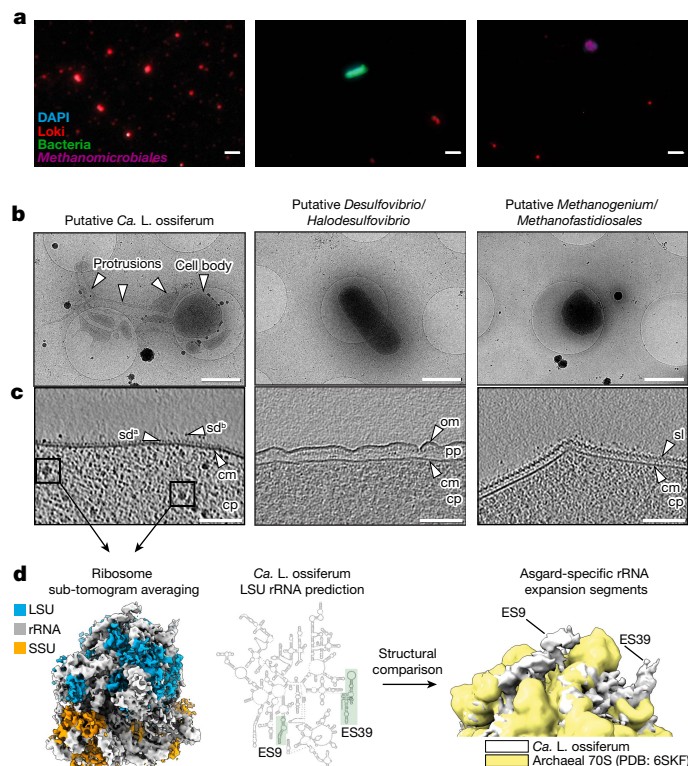

**Fig. 3 | Identification of 'Ca. L. ossiferum' cells in the enrichment culture.**
**a**, Hybridization chain reaction-FISH analysis of the enrichment culture stained with DAPI (cyan) and nucleotide probes targeting the major species of the culture, that is, Lokiarchaea cells (red; the sample on the left was 70× concentrated), bacteria (green) and *Methanomicrobiales* (purple). The FISH experiments were performed five independent times with similar results. Scale bars, 2 µm. **b**, Low-magnification 2D cryo-electron micrographs of the three major cell types that were observed after screening of the enrichment culture (*n* = 2 independent cultures), showing a putative 'Ca. L. ossiferum' cell with a round cell body and complex cell protrusions (left), a Gram-negative bacterial cell (middle) and an archaeal cell (right). Scale bars, 1 µm. **c**, Slices through cryo-tomograms of all three organisms shown in **c** (slice thickness, 9.02 nm), detailing the characteristic cell envelope architecture of the three species. Putative Lokiarchaea show small and unordered surface densities (sd[a]) and complex surface proteins (sd[b]) protruding from a single membrane. cm, cytoplasmic membrane; cp, cytoplasm; om, outer membrane; pp, periplasm; sl, surface layer. Scale bars, 100 nm. **d**, Identification of 'Ca. L. ossiferum' by Asgard-specific rRNA structures. Left, a sub-tomogram average (11.7 Å resolution) of ribosomes from cryo-tomograms of putative lokiarchaeal cells (large-subunit proteins (LSU), blue; small-subunit proteins (SSU), orange; rRNA, white). Middle, secondary structure prediction of the 'Ca. L. ossiferum' large-subunit rRNA (expansion segments ES9/ES39 are labelled). Right, a superposition of the average with a low-pass filtered (11 Å) map of the *T. kodakarensis* 70S ribosome (Protein Data Bank (PDB): 6SKF; yellow). The 'Ca. L. ossiferum' structure (white) shows prominent additional rRNA features that were identified as the Asgard-specific rRNA expansion segments ES9 and ES39. See also Extended Data Fig. 6.

cell bodies associated with elaborate and heterogeneous protrusions. The other two cell types were rods and spherically shaped cells, respectively, without protrusions. Their cell envelopes had canonical Gram-negative and archaeal cell envelope features, therefore probably representing co-cultured bacteria and archaea (Fig. 3c). By contrast, the cell envelope of 'Ca. L. ossiferum' candidates featured complex unordered densities protruding from a single membrane.

To unambiguously identify 'Ca. L. ossiferum' cells, we initially attempted to correlate FISH with cryo-ET. FISH, however, involves harsh sample preparation steps (chemical fixation, dehydration, permeabilization and high temperatures), which did not allow for the preservation of the fragile cellular ultrastructure. We therefore turned to an alternative approach and hypothesized that 'Ca. L. ossiferum' cells could be identified by unique structural features of their ribosomes. Supersized eukaryote-like expansion segments (ES9 and ES39) have been reported in Asgard archaeal rRNAs but are absent from bacteria and other archaea[36]. We therefore computationally extracted 4,126 sub-tomographic volumes of ribosomes from 56 tomograms of 'Ca. L. ossiferum' candidate cell bodies and protrusions and averaged them (Fig. 3d). A comparison between the average and a high-resolution structure of a euryarchaeotal 70S ribosome revealed additional prominent densities, which we identified as Asgard-specific expansion segments ES9 and ES39, by correlating large subunit rRNA positions to the structural docking result (Fig. 3d and Extended Data Fig. 6c). Importantly, expansion segments were not present in the large-subunit rRNA sequences of co-cultured species (Extended Data Fig. 6d).

The cell type exhibiting a characteristic variable morphology, cell bodies with long protrusions and a single membrane with an elaborate surface proteome was therefore identified as 'Ca. L. ossiferum'. These cells often appeared as individuals in FISH analyses (Extended Data Fig. 7a) and electron microscopy imaging (Fig. 3b). However, sometimes, 'Ca. L. ossiferum' was found in aggregates with co-cultured species (Extended Data Fig. 7b,c). As these observations were rather infrequent, we assumed that, although nutrient exchange with co-cultured species may be necessary for the strain's growth, persistent cell–cell contact seems to not be obligate throughout its life cycle.

## Complex and variable cell architecture

Having established the identity and general appearance of 'Ca. L. ossiferum' cells, we next aimed to analyse their overall organization by scanning EM. We identified small coccoid cells with surface-bound vesicles and extensive protrusions (Fig. 4a,b and Extended Data Fig. 7d). In contrast to 'Ca. P. syntrophicum', these long protrusions appeared more irregular, frequently branching or expanding into bulbous structures.

To investigate the macromolecular organization of 'Ca. L. ossiferum' cells, we extended our cryo-ET analysis. In contrast to data obtained using scanning electron microscopy (SEM), membranous protrusions of unfixed cells showed even more elaborate shapes (Extended Data Fig. 7e). Some protrusions connected multiple larger cell bodies (Fig. 4c (inset)), reminiscent of cell bridges observed in other archaea[37–40]. Although the majority of cells exceeded the thickness limitation of cryo-ET, some were thin enough to visualize the internal architecture (Fig. 4c) in 3D, enabling us to not only observe large protein structures such as ribosomes, but also numerous thin and sometimes bent filaments (convolutional neural network-aided segmentations are shown in Fig. 4d,f; Supplementary Video 1). This network of filaments was even better resolved in some of the very thin (<100 nm) protrusions (Fig. 4e,f and Supplementary Video 1), which contained sometimes bundled filaments that connected different parts of the cell. Ribosomes were homogenously distributed throughout the cell body, cell bridges and protrusions (Fig. 4c–f), where they could sometimes be observed as membrane-associated chains (Fig. 4g). Except for two instances (Extended Data Fig. 7f), we did not observe internal membrane-bound compartments.

Our extended cryo-ET dataset also revealed further insights into the structure of the cell envelope. The single membrane was not only decorated with a layer of small unordered densities, but also with a plethora of structures further protruding from the membrane (Fig. 4c–l (blue arrowheads)). Some densities connected different parts of the protrusion network (Fig. 4d,f), whereas others formed elaborate assemblies that localized to regions of high membrane curvature (Fig. 4f,h). On the cytoplasmic side of the membrane, we infrequently (*n* = 2) observed putative chemoreceptor arrays (Fig. 4i). Consistent with this observation, the genome of 'Ca. L. ossiferum' encodes a set of chemotaxis proteins. Although absent from 'Ca. P. syntrophicum', many of these are

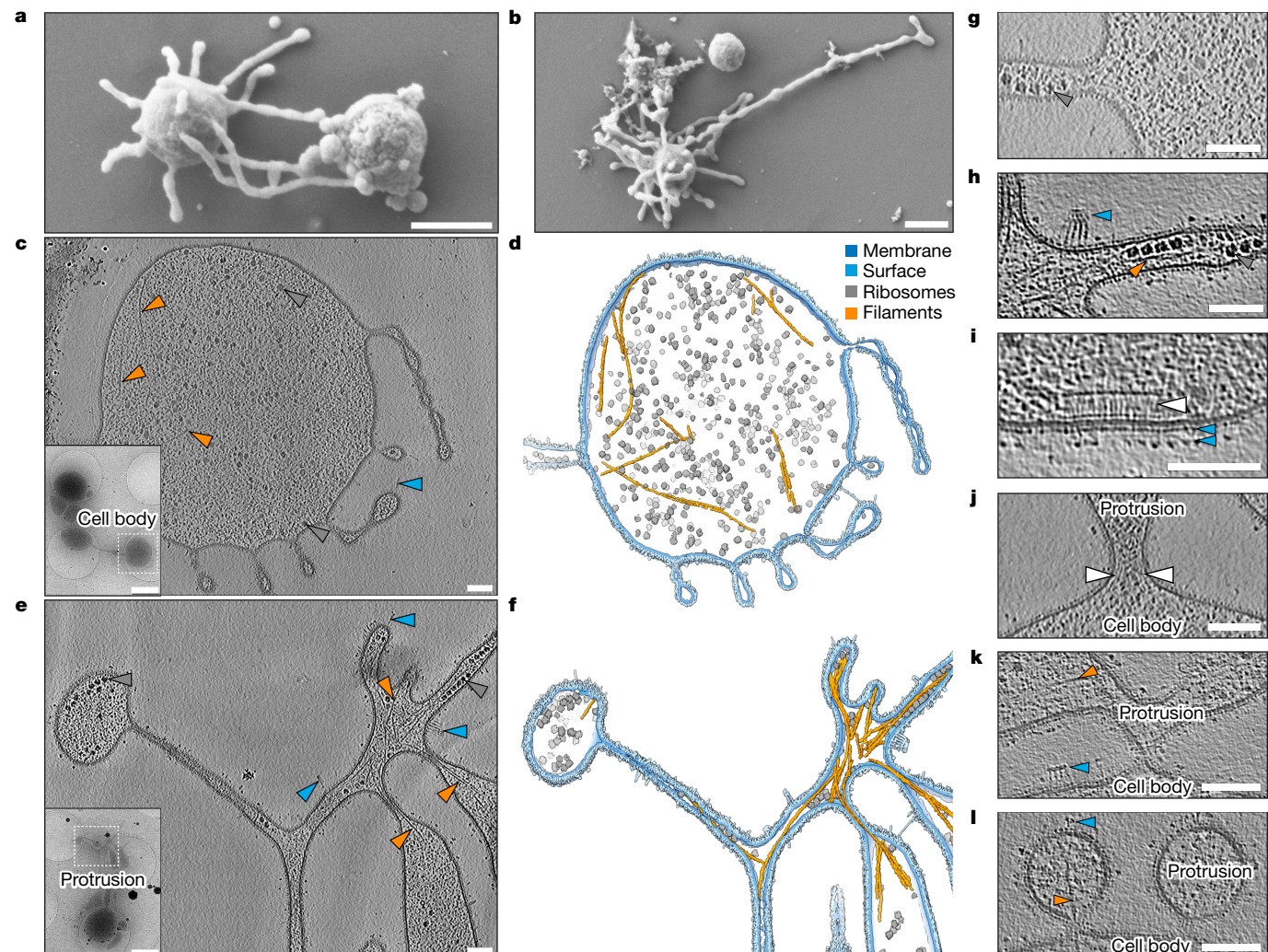

**Fig. 4 | Complex and variable architecture of 'Ca. L. ossiferum' cells.**
**a**,**b**, SEM imaging of fixed 'Ca. L. ossiferum' cells showed small coccoid cells with extensive protrusions. Example micrographs from n = 2 independent cultures are shown. See also Extended Data Fig. 7d. For **a** and **b**, scale bars, 500 nm. **c**–**f**, Slices through cryo-tomograms (**c**,**e**; thickness, 9.02 nm) and the corresponding neural-network-aided 3D volume segmentations (**d**,**f**) of two different 'Ca. L. ossiferum' cells. The insets in **c** and **e** show 2D overview images of the two different target cells. Cell bodies (**c**,**d**) and networks of protrusions (**e**,**f**) both contained ribosomes (grey arrowheads), cytoskeletal filaments (orange arrowheads) and complex surface densities (blue arrowheads). Note that **e** and **f** show the same cell as in Fig. 3c. For **c** and **e**, scale bars, 100 nm

(tomogram) and 1 μm (2D overview). **g**,**h**, Expanded views of slices from tomograms in **c** and **e**, showing ribosome chains, complex surface proteins and filaments (colour code as in **c**–**f**) in a junction of a cell bridge (**g**) and a constricted part of the protrusion network (**h**). For **g** and **h**, scale bars, 100 nm. **i**–**l**, Slices through cryo-tomograms showing a putative chemoreceptor array (**i**; indicated by a white arrowhead) and different types of connections between cell bodies and protrusions (**j**–**l**). The coloured arrowheads indicate filaments and surface structures as defined for **c**–**f**. The white arrowheads in **j** indicate weak densities at the neck of the junction. Slice thickness, 9.02 nm (**j**) or 10.71 nm (**i** and **k**–**l**). For **i**–**l**, scale bars, 100 nm.

also present in other Lokiarchaeia and Heimdallarchaeia (in 27 out of 97 MAGs; Extended Data Fig. 8). The gene set includes 15 methyl-accepting chemotaxis proteins as well as *cheA/B/C/D/R/W/Y* (Supplementary Table 4). Together with the extensive repertoire of surface proteins, these may mediate cell–cell communication, interactions and motility.

The unique cell envelope was mostly continuous between the cell body and protrusions, even though the transition zones showed high variability. Some appeared as stable junctions (Fig. 4j) (with potential densities appearing to stabilize the 'neck'), whereas others formed very thin constrictions (Fig. 4c,d) or were only loosely attached (Fig. 4k,l). Notably, single cytoskeletal filaments frequently traversed the junctions into the protrusions (Fig. 4l). In a similar manner, filaments also connected different parts of the protrusion network, often extending across constricted membrane tubes (Fig. 4f). These observations indicate that the cytoskeleton functions as a scaffold to maintain the elaborate cellular architecture of 'Ca. L. ossiferum'.

## Lokiactin-based cytoskeleton

To resolve the identity of the most frequently observed cytoskeletal filament in cryo-tomograms (Figs. 4 and 5a), we set out to determine its structure in situ and developed a workflow using helical reconstruction of 2D-projected sub-tomograms, followed by sub-tomogram averaging (Extended Data Fig. 9). 2D classification showed a two-stranded filament structure (Fig. 5b). The final reconstruction at a resolution of 24.5 Å enabled us to determine the helical parameters (rise of 27.9 Å

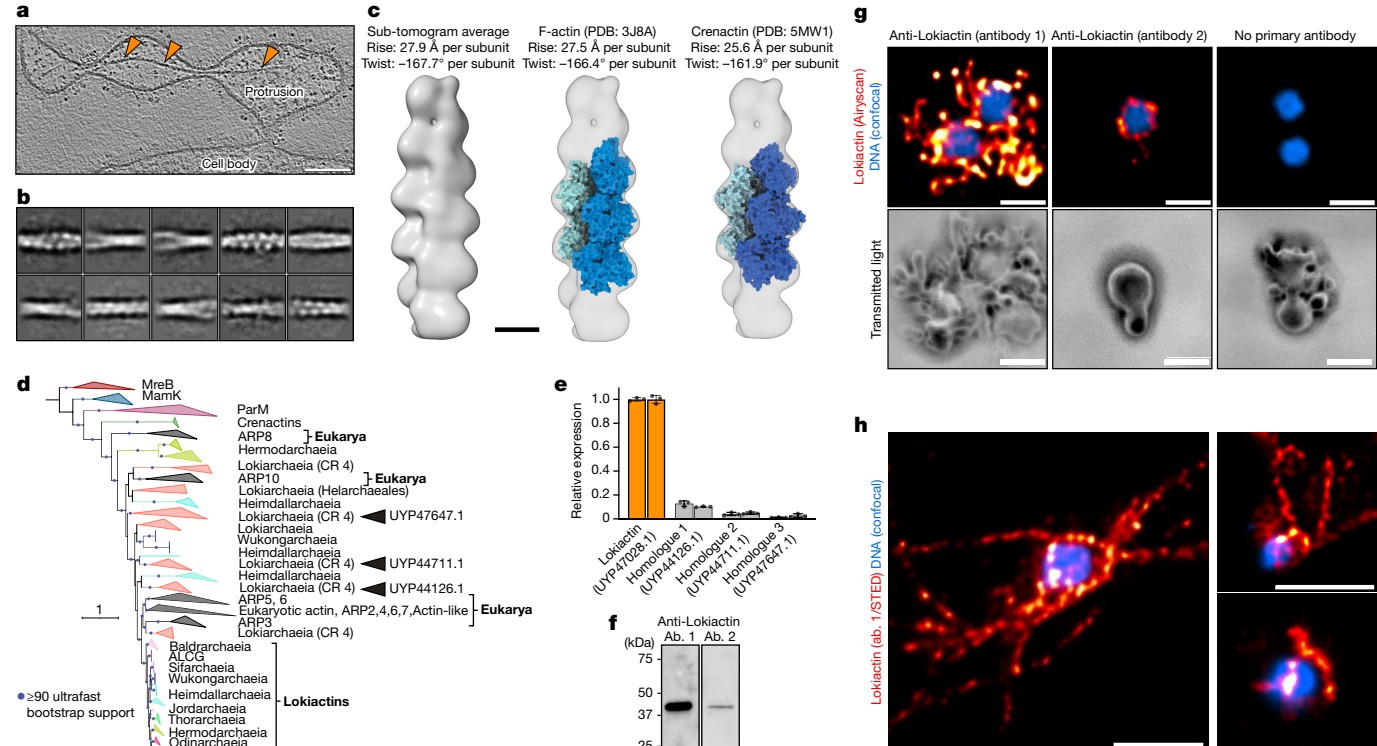

**Fig. 5 | Lokiactin is involved in cytoskeleton formation. a**, Slice through a cryo-tomogram showing a cytoskeletal filament inside a protrusion at higher magnification. Slice thickness, 5.36 nm. Scale bar, 100 nm. **b**, Filament segments were extracted from cryo-tomograms for structural analysis. 2D classes that were obtained after helical reconstruction of 2D-projected filament particles are shown, indicating a twisted double-stranded architecture. Box size, 34.3 × 34.3 nm. See also Extended Data Fig. 9. **c**, Sub-tomogram average (24.5 Å resolution) of the cytoskeletal filament displaying helical parameters with a high similarity to eukaryotic F-actin and archaeal Crenactin. Structural docking shows that an F-actin-like filament is consistent with the reconstructed map. See also Extended Data Fig. 9. Scale bar, 50 Å. **d**, Maximum-likelihood phylogenetic tree of actin family proteins. The '*Ca.* L. ossiferum' genome encodes four homologues. One homologue (GenBank: UYP47028.1) clusters together with other Asgard archaeal Lokiactins (group indicated by the orange arrowhead), which form a sister group to eukaryotic actin. The three other homologues (from top to bottom: GenBank UYP47647.1, UYP44711.1 and UYP44126.1) cluster with other Asgard archaeal and eukaryotic ARPs (groups indicated by the black arrowheads). The tree was rooted with the MreB protein family. CR 4, subgroup from within Lokiarchaeia. **e,f**, Lokiactin is expressed in the enrichment culture. **e**, Transcription of the four actin homologues was analysed using RT–qPCR analysis of two enrichment cultures, indicating the highest levels of transcripts per subunit, twist of −167.7 Å per subunit), which are highly similar to for Lokiactin. The mean expression levels normalized to Lokiactin are shown. The error bars indicate the s.d. of three technical replicates. **f**, Expression was also detected using western blotting analysis of a cell lysate obtained from the enrichment culture (representative result from $n = 2$ independent samples). Gel source data are provided in Supplementary Fig. 1. Two antibodies (ab.) were used that were raised against different '*Ca.* L. ossiferum' Lokiactin-specific peptides. **g,h**, Immunofluorescence staining of '*Ca.* L. ossiferum' cells with two different Lokiactin-specific antibodies analysed using Airyscan (**g**) or stimulated emission depletion (STED) (**h**) imaging (representative images of $n = 3$ (**g**) or $n = 2$ (**h**) independent preparations). The distribution of fluorescent signal indicates the presence of Lokiactin-based cytoskeletal structures in cell bodies and protrusions, being consistent with observations from cryo-tomograms. The top row of **g** shows single slices of the fluorescent DNA signal (blue, Hoechst stain, LSM-Plus-processed confocal) and the Alexa Fluor 647-labelled secondary antibodies (red/orange, jDCV-processed Airyscan). The bottom row of **g** shows the minimum intensity z-projections of the transmitted light channel to indicate the cell shape. The control (right column) was probed with only secondary antibodies (the contrast in the top row was adjusted equally). The images in **h** show single slices of representative deconvolved STED images detecting Lokiactin (red/orange, abberior STAR 580-labelled secondary antibodies) and DNA (blue, SPY505-DNA). For **g** and **h**, scale bars, 1 μm.

per subunit, twist of −167.7 Å per subunit), which are highly similar to eukaryotic F-actin but also to the archaeal Crenactin[23]. Importantly, the dimensions of F-actin[41] and Crenactin[23] filaments would be consistent with our reconstructed map (Fig. 5c).

The '*Ca.* L. ossiferum' genome contains four homologues of eukaryotic actin. One of these homologues clusters inside a group of Lokiactins, which form a sister group to bona fide eukaryotic actin and most ARPs in phylogenies (Fig. 5d). The remaining three homologues cluster with other ARPs from eukaryotes and Asgard archaea. Their phylogenetic patterns and distribution indicate a complex evolution within Asgard archaea, which seems to be shaped by gene losses, gains and duplications. By contrast, Lokiactin represents the most highly conserved group of actin homologues. It is found in all Asgard lineages[5], implicating its presence in the last common ancestor of the entire phylum (Fig. 5d).

To test whether the actin-like cytoskeletal filament in '*Ca.* L. ossiferum' contained Lokiactin, we first tested the expression levels of all actin homologues using qPCR with reverse transcription (RT–qPCR) and found that Lokiactin expression was severalfold higher (Fig. 5e). We next generated two antibodies that were raised against Lokiactin-specific peptides. Using these antibodies, expression was detected in lokiarchaeal cell lysates by western blotting (Fig. 5f) and we observed staining throughout '*Ca.* L. ossiferum' cells in immunogold-labelling experiments (Extended Data Fig. 10). In immunofluorescence experiments, both antibodies revealed filamentous signals in the cell bodies and particularly also in protrusions, consistent with the abundant distribution of filaments observed in cryo-tomograms (Fig 5g,h). We therefore conclude that '*Ca.* L. ossiferum' possesses a complex Lokiactin-based cytoskeleton. As '*Ca.* L. ossiferum' encodes eight gelsolin-like and three profilin-like proteins that have been show to affect actin polymerization

dynamics[4,5,19,24,25], we hypothesize that the assembly of the complex Lokiactin cytoskeleton in 'Ca. L. ossiferum' is probably dynamically regulated by actin-binding proteins.

## Conclusion

In conclusion, our comparatively fast-growing lokiarchaeal culture enabled us to study its cell architecture in a near-native state. We discovered an elaborate actin-based cytoskeleton in Asgard archaea, which has long been hypothesized[4–6,42–44], but has not been visualized.

The cytoskeleton is probably a hallmark structure of all Asgard archaea, as Lokiactin is conserved in genomes of all Asgardarchaeota classes. This clearly differentiates Lokiactin from many other ESPs that show complex evolutionary histories of gene gains and losses and lateral transfer resulting in patchy distributions in Asgard and also other archaea[2]. The high degree of Lokiactin conservation indicates strong constraints on its function. It is therefore probable that a dynamic cytoskeleton, regulated by numerous actin-related and actin-binding proteins, had a substantial impact on the emergence, evolution and diversification of Asgard archaea. Our cryo-ET and immunofluorescence data revealed actin filaments in both parts of the cell. In the cell body, filaments are often found at the periphery, and they follow the longitudinal axis in the cell's protrusions. We therefore propose that Lokiactin acts as a scaffold for the complicated cell architecture of Asgard archaea, similar to eukaryotic actin, which is a major determinant of eukaryotic cell shape[45].

The elaborate cell architecture with extensive membranous protrusions has multiple implications for Asgard physiology and ecology. As these characteristic features make the cells highly fragile, it could also explain why the highest abundance of Asgard archaea is found in sediments[30,46–48] rather than in plankton. Importantly, our study established approaches that will enable imaging of Asgard archaea in a culture-independent manner in environmental samples, with the possibility of identifying Asgard cells in cryo-tomograms based on unique ribosomal RNA expansion segments.

The large surface area of the convoluted network of protrusions, in combination with the unusual cell envelope lacking a highly ordered S-layer (as typically found in other archaea) but rather displaying numerous surface proteins, may have enabled the intricate cell–cell contacts required for eukaryogenesis that—considering the lifestyle of the two cultured Asgard strains—probably involved interspecies dependencies in syntrophic relationships[49–52]. These findings strongly support a gradual path of mitochondrial acquisition through protrusion-mediated cell–cell interactions, which have been proposed previously in the inside-out and E3 hypotheses[7,28]. Additional experimental data—in particular, from diverse Asgard archaea—will be needed to further test these models and exclude alternative views[42,43]. Importantly, the cell architecture may enable the compartmentalization of cellular processes even in the absence of the internal organelle-like membrane systems that had been hypothesized based on the genomes of the first Asgard archaea[10].

Finally, our enrichment culture will serve as a model system to study the peculiar cell biology of Asgard archaea, as it grows to cell densities and purities that make it experimentally tractable. The intricate cell architecture of 'Ca. L. ossiferum' suggests elaborate mechanisms for processes such as macromolecular trafficking and assembly, membrane shaping, cell division and spatiotemporal regulation of the cytoskeleton, many of which are probably mediated by ESPs that can now be studied in situ.

**Etymology.** *Lokiarchaeum* (Lo.ki.ar.chae'um. N.L. neut. n. *archaeum* (from Gr. masc. adj. *archaios*, ancient), an archaeon; N.L. neut. n. *Lokiarchaeum*, an archaeon named after Loki, a god in Norse mythology). *ossiferum* (os.si'fe.rum. L. neut. pl. n. *ossa*, skeleton; L. v. *fero*, to carry; N.L. neut. adj. *ossiferum*, skeleton-carrying). The name describes a skeleton-carrying archaeon of the provisional class Lokiarchaeia[11] within the Asgardarchaeota phylum.

**Locality.** Isolated from a shallow sediment of an estuarine canal in Piran, Slovenia.

**Diagnosis.** Anaerobic archaeon of the Asgardarchaeota phylum that grows in enrichments with $H_2$-consuming bacteria and archaea at 20 °C on organic medium. Maximal cell densities reach $5 \times 10^7$ cells per ml at relative enrichments of up to 79%. Cells grow extensive, often branched protrusions with blebs and exhibit complex, irregular surface structures. The genome is 1,607,517 bp larger than that of the closest relative 'Ca. P. syntrophicum' and shows 58.4% similarity at the amino acid level.

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

## Methods

### Sample collection and enrichment cultivation

Sediment core samples (length, approximately 60 cm) were retrieved from a shallow brackish canal (at 40 cm water depth, 19.5 °C, pH 8) near the coast of Piran, Slovenia (45° 29′ 46.1″ N 13° 36′ 10.0″ E) on 21 April 2019.

The sediment core was cut inside an anaerobic tent ($N_2$ atmosphere) at intervals of 3 cm, with each fraction being placed inside 50 ml conical sterile centrifuge tubes, sealed and stored at 4 °C. Around 0.5 g of sediment from the different fractions was used for DNA extraction and subsequent qPCR assays targeting lokiarchaeal 16S rRNA genes (qPCR conditions are provided below). The 13–16 cm deep sediment layer with the highest relative 16S rRNA gene copies of Lokiarchaea per ml in relation to total DNA content (Extended Data Fig. 2) was selected for further use in cultivation, which was performed in 120 ml serum bottles sealed with butyl rubber stoppers. A total of 2 g of sediment was used as inoculum and different medium and headspace conditions were tested; growth was monitored using qPCR assays as described above. After 140 days, the growth was observed in cultures inoculated in 50 ml of sterile-filtered brackish water from the canal supplemented with casein hydrolysate (0.1% (w/v)), 20 amino acids (0.1 mM each) and milk powder (0.1%) incubated at 20 °C under an atmosphere of 80:20 $N_2$:$CO_2$ (0.3 bar). To try to limit bacterial growth, cultures were also supplemented with ampicillin, kanamycin and streptomycin (50 µg ml$^{-1}$ each). The cultures were transferred into fresh medium whenever exponential growth could be detected (between 1–3 months). After two transfers, growth could no longer be observed under this set-up and the next transfer was performed in modified '*Ca*. P. syntrophicum' MK-D1 medium[7]. After three transfers under these conditions, the medium was reduced to further limit bacterial growth and, after five additional transfers in minimal medium, high enrichments were achieved. Eventually, the growth medium composition (per litre) was as follows: 20.7 g NaCl, 5 g $MgCl_2$·$6H_2O$, 2.7 g $NaHCO_3$, 1.36 g $CaCl_2$·$2H_2O$, 0.54 g $NH_4Cl$, 0.14 g $KH_2PO_4$, 0.03 g $Na_2S$·$9H_2O$, 0.03 g cysteine·HCl, 0.5 ml of acid trace element solution, 0.5 ml of alkaline trace element solution, 1 ml Se/W solution, 0.1% casein hydrolysate (w/v). The acid trace element solution contained (per litre): 1.491 g $FeCl_2$·$4H_2O$, 0.062 g $H_3BO_3$, 0.068 g $ZnCl_2$, 0.017 g $CuCl_2$·$H_2O$, 0.099 g $MnCl_2$·$4H_2O$, 0.119 g $CoCl_2$·$6H_2O$, 0.024 g $NiCl_2$·$6H_2O$ and 4.106 ml of HCl (37%). The alkaline trace element solution contained (per litre): 0.017 g $Na_2SeO_3$, 0.033 g $Na_2WO_4$, 0.021 g $Na_2MoO_4$ and 0.4 g NaOH. The medium pH was adjusted to 7.5 and contained ampicillin, kanamycin and streptomycin (200 µg ml$^{-1}$ each). The headspace atmosphere was 80:20 $N_2$:$CO_2$ (0.3 bar) and the cultures were incubated at 20 °C without shaking.

### DNA extraction and growth monitoring using qPCR

A total of 2 ml of the cultures was sampled every 14 days and centrifuged for 30 min at 20,000*g* at 4 °C. The supernatant was discarded and the resulting pellet was resuspended in 700 µl of SL1 buffer from the NucleoSpin Soil DNA extraction kit (Macherey-Nagel). The rest of the procedure was performed according to the manufacturer's instructions. High molecular mass DNA for genome sequencing was extracted using a standard phenol–chloroform-based protocol. The DNA concentration was measured with the Qubit 2.0 Fluorometer (Invitrogen), using the dsDNA HS kit, according to the manufacturer's instructions.

Lokiarchaea-specific 16S rRNA gene primers (LkF, 5′-ATCGAT AGGGGCCGTGAGAG and LkR, 5′-CCCGACCACTTGAAGAGCTG) were designed using the ARB tool[54]. All assays were performed in triplicates on the CFX Connect Real-Time PCR Detection System (Bio-Rad) and data were collected using the CFX Maestro (v.2.3). Reaction mixtures (20 µl) contained: 1× Luna Universal qPCR Master Mix (New England BioLabs), 0.5 µM of each primer and 5–10 ng of template DNA. The cycling conditions were as follows: 95 °C for 1 min; then 40 cycles of 15 s at 95 °C and 1 min at 60 °C (for both annealing and extension), with a fluorescence reading after each cycle. Melting curves were generated by increasing the temperature from 60 °C to 95 °C, at 0.5 °C increments for 5 s with fluorescence readings after each increment. Lokiarchaeal 16S rRNA gene fragments amplified from sediment DNA were used as quantification standards. For quantification, triplicates of standard tenfold dilutions ranging from 10 to 10$^8$ copies were used in every assay. The efficiencies of these reactions varied from 90% to 100%, with $R^2$ values of >0.99. Primer specificity was confirmed through amplicon sequencing (Illumina MiSeq) of environmental DNA (Extended Data Fig. 3).

### RNA extraction, cDNA synthesis and ARP RT–qPCR

From cultures, 20 ml was centrifuged at 20,000*g* for 30 min, 4 °C. The pellet was then resuspended in 600 µl of the lysis/binding buffer from the mirVana miRNA isolation kit (Invitrogen). The rest of the procedure was performed according to the manufacturer's instructions. Potential leftover genomic DNA was removed by incubating the samples with TURBO DNase (Invitrogen) at 37 °C for 1 h. Lokiarchaeal 16S rRNA PCR tests were then used to confirm that no DNA had remained in the sample. cDNA was produced using the ProtoScript II First Strand cDNA Synthesis Kit according to the manufacturer's instructions.

Primers targeting Lokiactin (GenBank: UYP47028.1, 83F, 5′-GCAGGAGAAGATCAGCCTCG; 337R, 5′-AACCGGATGTTCGCTTGGAT) and actin homologue (GenBank: UYP44126.1; 82F, 5′-TGGGGGAGA AAATGAGCCAC; 443R, 5′-GGCCCCACGAACAGGATAAT), GenBank UYP44711.1 (361F, 5′-CCCTCCCAGACATTGCACAA; 731R, 5′-TGCG GGATCGACAGAATCAG) and GenBank UYP47647.1 (305F, 5′-CAAGGC TGGATCCCTTCAGA; 591R, 5′-ATTGCGTGATATGGTGGCCT) were designed using the Geneious Prime 2021.2 software (https://www. geneious.com). A temperature gradient PCR using culture DNA was used to determine the optimal annealing temperature and the temperature used for all primer pairs was 60 °C. The qPCR procedure and cycling conditions were the same as those used for lokiarchaeal 16S rRNA gene amplification. Amplified fragments from culture DNA were used as standards. The efficiencies of these reactions varied from 90% to 100%, with $R^2$ values of >0.99. Primer specificity was evaluated by melting curve analysis.

### 16S rRNA gene amplicon sequencing

Amplicon sequencing was performed by amplifying the extracted DNA using the general prokaryotic 16S rRNA gene targeting primer pair 515f (5′-GTGCCAGCMGCCGCGGTAA) and 806r (5′-GGACTACHVGGGTWTCTAAT)[55], which was then barcoded and sequenced at the Vienna BioCenter Core Facilities (VBCF) using the Illumina Miseq (300 PE) platform. Raw reads were processed using cutadapt[56] to remove primer sequences followed by the sequence analyses using the QIIME2 pipeline[57]. In brief, the DADA2 algorithm was used to denoise the data as well as to remove low-quality reads and chimeras. Sequences with 100% sequence identity were clustered into amplicon sequence variants. Taxonomy of amplicon sequence variants was assigned using the SILVA database (release 138) with the q2-feature-classifier plugin[58].

### In situ DNA-hybridization chain reaction

Cells were fixed in growth medium with the addition of 2.5% formaldehyde for 2 h at room temperature. After this period, they were washed three times in 1× phosphate-buffered saline and then stored at −20 °C in a mixture of absolute ethanol and 1× phosphate-buffered saline (1:1). The oligonucleotide probes used in this study were the general bacteria-targeting EUB338 (ref. [59]), the Methanomicrobiales-targeting MG1200 (ref. [60]) and the Lokiarchaea-specific DSAG-Gr2-1142 (ref. [7]) (Supplementary Table 5). The DNA-hybridization chain reaction procedure was performed as described previously[61] and the samples were imaged on the Eclipse Ni-U epifluorescence microscope (Nikon) using Gryphax (v.1.1.8.153). Images were processed using ImageJ[62].

## Illumina sequencing

For short-read sequencing, metagenomic DNA of three samples from different time points in our enrichment was shotgun sequenced using the NovaSeq 6000 (paired-end, 150 bp) platform at Novogene. Sequencing data were processed using Trimmomatic (v.0.36)[63] to remove Illumina adapters and low-quality reads (SLIDINGWINDOW:5:20). The trimmed reads were co-assembled using SPAdes (v.3.15.2)[64] with the *k*-mer length varying from 21 to 111 and the '--meta' option. Contigs longer than 1,000 bp were binned using Binsanity[65], MaxBin2 (ref. [66]), MetaBAT[67] and CONCOCT[68] followed by contig dereplication and binning optimization using the DAS tool[69]. Completeness and contamination of MAGs was evaluated using CheckM[53] and taxonomic affiliation was obtained using GTDB-Tk (v.1.5.0; classify_wf)[70]. A lokiarchaeal MAG (genome length, 6,008,683 bp; completeness, 90.9%; contamination, 7.9%) was identified.

## Nanopore sequencing and base calling

Library construction for ONT sequencing was performed using the SQK-LSK109 ligation kit followed by the PromethION sequencing using FLO-PRO002 flowcells. Fast5 files were processed using the high-accuracy ONT basecaller Bonito (v.0.3.6; https://github.com/nanoporetech/bonito) with the dna_r9.4.1 pretrained model (bonito basecaller --fastq dna_r9.4.1). Long-read demultiplexing and adapter removal was performed using Porechop (v.0.2.4)[71]. Reads shorter than 1 kb were removed using NanoFilt (v.2.8.0; NanoFilt -l 1000)[72]. We performed an initial metagenomic assembly of long reads using Flye (v.2.8.3-b1695)[73] with the '--meta' option. On the basis of the taxonomic assignment of contigs using the GTDB-Tk (v.1.5.0) classify workflow (classify_wf)[70], we identified a Lokiarchaeal genome with a total length of 6,035,157 bp in a single non-circular contig. To obtain the complete Lokiarchaeal genome, the following approach was applied. Long reads mapping to the short-read-assembled MAG and the long-read-assembled non-circular contig were extracted using minimap2 (ref. [74]) with the '-ax map-ont' option. Mapped reads were converted to BAM format using samtools[75], and bedtools bamtofastq[76] was used to obtain the reads in FASTQ format. Duplicated sequences were removed using the SeqKit[77] tool. After long-read deduplication, the remaining reads were assembled using flye (--nano-raw)[78] yielding a circular genome. Polishing and validation of the Loki-B35 genome was performed using four rounds of Pilon[79] as part of the validation pipeline of metagenomic assemblies proposed previously[80].

## Phylogenomic tree

We recovered a total of 167 good-quality Asgard archaea genomes (completeness over 80% and less than 10% contamination) from the NCBI and other public databases (Supplementary Table 6). To reconstruct the phylogenomic tree of life, we collected a total of 23 ribosomal protein markers (Supplementary Table 7) from a wide range of bacteria, non-Asgard archaea and eukaryotes (Supplementary Tables 8 and 9) from a previous study[2].

The identification and retrieval of the 23 ribosomal markers in our Asgard genome database was based on the proteome annotation performed using Prokka (v.1.14.6)[81]. Two phylogenomic trees were reconstructed in this study: (1) a tree of life with ribosomal markers from bacteria, eukaryotes, non-Asgard archaea and a subset of 94 Asgard genomes (Supplementary Table 6); and (2) a second tree using sequences only from Thermoproteota and the 168 Asgardarchaeota genomes present in our database. Ribosomal markers were aligned using the L-INSi algorithm of MAFFT (v.7.427)[82] and trimmed using BMGE using the default parameters[83]. Both sets of ribosomal markers were concatenated independently resulting in two multiple sequence alignment comprising 2,327 and 3,285 amino acid sites for the tree of life and the Thermoproteota plus Asgardarchaeota datasets, respectively. Phylogenomic reconstructions were performed with IQ-TREE 2 (ref. [84])

under the LG + C20 + F + G model with 1,000 ultrafast bootstrap replicates using UFBoot2 (ref. [85]).

## Genome analysis and creation of orthogroups between 'Ca. L. ossiferum' and 'Ca. P. syntrophicum'

Protein sequences of 'Ca. L. ossiferum' and 'Ca. P. syntrophicum' were predicted using Prokka (v.1.14.6)[81]. The identified proteins were annotated using HMM searches against the Pfam database (v.34.0)[86] and KEGG Orthology was assigned using the BlastKOALA online server[87].

Proteomes of 'Ca. L. ossiferum' and 'Ca. P. syntrophicum' were used as queries in BLASTp searches ($e = 1 \times 10^{-10}$) against the recently published collection of Asgard clusters of orthologous genes[2] to obtain the asCOG numbers and annotation. Best matches were selected according to the following criteria: asCOG annotation of best hit was assigned if the BLAST alignment covered more than 70% of the query and sequence identity was greater than 25%; when the best BLAST hits did not meet the 70% threshold, a lower query coverage of 30% and at least 25% sequence identity was used to annotate protein domains. The Asgard COG annotation was used to identify ESPs on the basis of the curated ESP set composed of 505 AsCOGs previously described[2]. Average amino acid identity between 'Ca. L. ossiferum' and 'Ca. P. syntrophicum' was calculated using CompareM (v.0.1.2) (with the 'aai_wf' parameter) (https://github.com/dparks1134/CompareM). Insertion sequence elements were identified using ISEScan[88].

Clusters of orthologous proteins between these two proteomes were identified using OrthoMCL (v.1.4)[89] according to the Synima pipeline[90] based on the all versus all protein alignments performed using BLASTp searches. Synima was also used to identify chains of syntenic genes between 'Ca. L. ossiferum' and 'Ca. P. syntrophicum' based on the DAGchainer software[91]. The genome synteny plot was constructed with Rideogram[92].

## Phylogeny of actin-family proteins

Actin-like proteins of Asgardarchaeota genomes were identified on the basis of BLASTp searches using Lokiactins[1] as queries against our curated set of Asgard genomes ($e = 1 \times 10^{-10}$, sequence identity ≥ 25%). To retrieve a list of eukaryotic actin sequences, proteins were downloaded from UniProt based on the accession IPR004000, which corresponds to the actin domain present in the actin family. Proteins were clustered using CD-HIT[93] (-c 0.99 -d 0 -g 1) to remove highly similar sequences. Lokiactins were used as queries using BLASTp and only sequences with more than 25% identity were retained. Eukaryotic actins and Asgard actin-like proteins were aligned together with bacterial actin homologues involved cell shape determination (MreB), magnetosome organization (MamK) and plasmid segregation (ParM) based on the structural information of homologues of the DASH database using MAFFT (v.7.490; --dash --localpair --originalseqonly) followed by a trimming step with trimAl (-gappyout)[94]. The phylogenetic tree was reconstructed using IQ-TREE 2.0 (ref. [82]) under the LG + R8 model selected by ModelFinder[95].

## SEM, conventional TEM and immunogold localization

For SEM, enrichment cultures of 'Ca. L. ossiferum' were prepared as described previously[96]. SEM was performed using the Zeiss Auriga field-emission scanning electron microscope (Zeiss), operated at 2 kV.

For conventional resin embedding of samples dedicated for transmission electron microscopy (TEM) or immunogold localization, the samples were high-pressure frozen and freeze-substituted using the Leica HPM100 and AFS2 systems, respectively (Leica). The freeze substitution medium consisted of ethanol containing 0.5% glutaraldehyde, 0.5% formaldehyde and 0.5% uranyl acetate. The freeze substitution program, the following embedding in Epon epoxy resin and immunogold localization with Lokiactin-specific primary antibodies (ab1, see below) were performed using the protocol described previously for *Phaeodactylum tricornutum*[97]. Immunogold controls for primary and

secondary antibody specificity were performed using *Vibrio harveyi* and *Pyrococcus furiosus*. Immunogold particle statistics were estimated for the controls as well as '*Ca*. L. ossiferum' (Supplementary Table 10). Transmission electron microscopy was performed using either the Zeiss EM 912 (Zeiss) system, operated at 80 kV and equipped with a Tröndle 2k × 2k slow-scan CCD camera (TRS, Tröndle Restlichtverstärker Systeme) or a JEOL F200 (JEOL), operated at 200 kV and equipped with a XAROSA 20 megapixel CMOS camera (EMSIS).

## Generation of Lokiactin-specific antibodies and western blotting
The Lokiactin antibodies were raised against Lokiactin-specific peptides (ab1: CTFYTDLRVDPSEHPV; ab2: CSKNGFAGEDQPRSVF) and validated through ELISA assays using the services of Eurogentec. Peptide antibodies were designed by Eurogentec.

For western blotting, an aliquot of the culture was centrifuged at 20,000*g* for 10 min at 4 °C, the pellet was washed once in base MLM medium without casein hydrolysate and lysed and denatured in SDS-loading buffer (Bio-Rad) containing 1% (v/v) beta-mercaptoethanol at 95 °C. The samples were run on 4–20% Tris-Glycine gradient gels (Bio-Rad), transferred to PVDF membranes, blocked with 5% milk powder in TBST and probed with primary and secondary (goat anti-rabbit HRP, Invitrogen, 31460) antibodies. Signals were detected by enhanced chemiluminescence (ECL).

## Immunofluorescence imaging
An aliquot of the culture was immobilized on poly-L-lysine-coated coverslips and fixed with 4% formaldehyde under a nitrogen atmosphere. Coverslips were blocked and permeabilized with 3% (w/v) BSA and 0.1% (v/v) Triton X-100 and subsequently probed with primary (anti-Lokiactin, 1:100 or 1:500 diluted) and secondary antibodies (either donkey anti-rabbit AF647, Invitrogen A-31573, 1:500 diluted; or goat anti-rabbit abberior STAR 580, abberior ST580-1002, 1:200 diluted) and counterstained with 10 μg ml⁻¹ Hoechst 33342 (Thermo Fisher Scientific, for Airyscan imaging) or SPY505-DNA (Spirochrome, for STED samples). Coverslips were mounted with Vectashield (Vector Laboratories, Airyscan samples) or Prolong Diamond (Thermo Fisher Scientific, STED samples). The samples were imaged using a Zeiss LSM900 with Airyscan 2 detector and a ×63/1.4 NA oil-immersion objective. *z*-stacks of target cells were recorded using one confocal imaging track detecting the Hoechst signal and transmitted light and a separate Airyscan track detecting the Alexa Fluor 647 signal. Confocal stacks were deconvolved using the Zeiss LSM Plus processing function and Airyscan images were processed with Zeiss joint deconvolution (jDCV, 15 iterations) in ZenBlue (v.3.5). Minimum-intensity projections of the transmitted light channel and extraction of single confocal/Airyscan slices were performed in Fiji[98]. STED images were acquired using a Leica SP8 STED equipped with a ×100/1.4 NA oil-immersion objective. DNA and Lokiactin signals were detected in stack-wise sequential imaging tracks. Images were deconvolved with Huygens Professional (v.22.04; Scientific Volume Imaging; http://svi.nl).

## Cryo-ET sample preparation
Samples were removed from the culture under a nitrogen atmosphere and mixed with 10 nm BSA-coated gold beads at a 1:5 ratio. The sample was kept in a nitrogen atmosphere until plunge-freezing. Then, 3.5 μl of the sample was applied to glow-discharged copper EM grids (R2/1, Quantifoil), automatically blotted from the backside (using a Teflon sheet on one side)[99] for 5–7 s and plunged into liquid ethane/propane[100] using the Vitrobot Mark IV (Thermo Fisher Scientific)[101].

## Cryo-ET data collection
Cryo-ET data were collected on a Titan Krios G4 (Thermo Fisher Scientific) system operating at 300 kV equipped with a BioContinuum imaging filter and a K3 direct electron detector (Gatan). Data acquisition was performed using SerialEM[102,103]. Owing to the low cell density of the non-concentrated sample, grids were first extensively screened using polygon montages at low magnification (×2,250). After identification of targets, tilt series were acquired using a dose-symmetric tilt scheme[104], covering an angular range of −60° to +60° and a total electron dose of 140–160 e⁻ Å⁻². Tilt series were either acquired at a pixel size of 4.51 Å at the specimen level using 2° angular increments between tilts and a target defocus of −8 μm or at higher magnification (pixel size of 2.68 Å) with 3° angular increments and a defocus ranging from −3 to −6 μm (for sub-tomogram averaging of the ribosome and reconstruction of the cytoskeletal filament). 2D projection images shown in the Article were recorded at a magnification of ×2,250 (pixel size of 39.05 Å) and a target defocus of −200 μm.

## Tomogram reconstruction, data processing and segmentation
Tilt series were drift-corrected using alignframes in IMOD[105] and 4×-binned tomograms were reconstructed by weighted-back projection in IMOD. To enhance the contrast for visualization and particle picking, tomograms were CTF-deconvolved and filtered using isonet[106]. 2D projection images were lowpass-filtered using mtffilter in IMOD. Segmentations were generated in Dragonfly (Object Research Systems, 2022; www.theobjects.com/dragonfly) as described previously[107]. In brief, isonet-filtered tomograms were further processed by histogram equalization and an unsharp filter, and a 5-class U-Net (with 2.5D input of 5 slices) was trained on 5–6 tomogram slices to recognize background voxels, filaments, membranes, cell surface structures and ribosomes. All neural-network-aided segmentations were cleaned up in Dragonfly, exported as a binary tiff and converted to mrc using tif2mrc in IMOD. Segmentations were visualized in ChimeraX[108].

## Sub-tomogram averaging of the ribosome
Sub-tomogram averaging of ribosomes was performed using RELION (v.4.0)[109]. The individual ribosome particles were manually picked using Dynamo[110] from 56 tomograms reconstructed at a binning factor of 4. The coordinates of particles and raw tilt series were imported into RELION (v.4.0) to generate pseudo sub-tomograms at a binning factor of 4 (4,126 particles). The particles were processed for 3D classification with a ribosome reference (Electron Microscopy Data Bank: EMDB-13448) that was low-pass filtered to 60 Å. The particles from good classes (class I and III in Extended Data Fig. 6) were processed for 3D auto-refinement at a binning factor of 2 and 3D local classification. The particles in the best 3D class (class II) were used for the 3D reconstruction at the binning factor of 1 and the resolution was further improved after three iterations of Tomo CTF refinement and frame alignment. The final structure of the ribosome at a resolution of 11.7 Å was reconstructed from 1,673 particles (Extended Data Fig. 6).

## Identification of '*Ca*. L. ossiferum' by rRNA expansion segments
The ribosome sub-tomogram average was compared to a high-resolution structure of a Euryarchaeota ribosome (PDB: 6SKF, *T. kodakarensis*[111]) using fitinmap in ChimeraX to identify unique rRNA features. For visualization, the high-resolution structure was lowpass-filtered to 11 Å resolution using molmap in ChimeraX. Large subunit rRNA secondary structures of '*Ca*. L. ossiferum' and *T. kodakarensis* were predicted using R2DT[112] (available at https://rnacentral.org/r2dt) to identify the position of Asgard-specific rRNA expansion segments[36] and positions were mapped to the structural docking result to define ES9 and ES39 in the sub-tomogram average.

## In situ reconstruction of cytoskeletal filaments
Filaments were analysed using a similar strategy as described in previous reports[113,114] and summarized in Extended Data Fig. 9a. In brief, individual filaments were manually picked from tomograms that were reconstructed at a binning factor of 4 using the filamentWithTorsion model in Dynamo. The filament segmentation was performed with an intersegment distance of 32.13 Å, resulting in a total of 10,031 segments

from CTF-corrected and dose-weighted tomograms at the binning factor of 2. These segmented subvolumes were processed for the filament analysis (Extended Data Fig. 9): (1) each subvolume was first rotated to make the helical axis parallel to the $z$ axis based on the orientation calculated from particle coordinates (or derived from Dynamo) and then further centred by alignment with a featureless cylinder-like reference. (2) Re-oriented subvolumes were further rotated 90° to orient particles parallel to the $xy$ plane. (3) The central slices of segments along the $z$ axis were extracted and projected to generate a 2D projection dataset. (4) 2D projection images were processed for RELION helical reconstruction analysis[115] with an actin reference (EMDB-11976) low-pass filtered to 60 Å. (5) The orientations of 2D projection images were mapped back into the corresponding subvolumes and the polarity of the filament was validated from the orientations of segments on the same filaments, while the c entres of segments were refined in 3D space by aligning against the 3D model from 2D projection images using relion_refine. The polarity voting results were not applied during the reconstruction, as the resolution was not sufficient for clear polarity determination. The segments with refined orientations were then processed for a second iteration of filament analysis, in which the reference in the RELION analysis was updated to the 3D model from 2D projection images.

After three iterations of filament analysis, a 2D classification without sampling was performed in the last iteration to select only particles in 2D classes with visible features. The particles were then processed for 3D helical refinement to optimize the helical parameters in the process. A total of 3,161 2D projections from subvolumes were used to determine a reconstructed map at a resolution of 31 Å with refined helical parameters of 28.56 Å (rise/subunit) and −168.50° (twist/subunit). To further improve the map quality, we next performed sub-tomogram averaging using RELION (v.4.0)[109] (Extended Data Fig. 9a). Individual filaments were manually picked from 56 tomograms in Dynamo and were segmented with an intersegment distance of 32.13 Å. The coordinates of 13,569 filament segments and original raw tilt series were imported into RELION (v.4.0) to generate pseudo sub-tomograms at a binning factor of 2. The particles were processed for 3D auto-refinement without imposing helical symmetry and were then applied to 3D classification without angular sampling. The particles in the good 3D classes (class I and IV) were selected for next round of 3D refinement, in which the helical parameters from the previously obtained 2D projection model were applied and optimized during refinement. The final structure at a resolution of 24.5 Å was reconstructed from 12,585 particles imposed with the refined helical parameters (rise = 27.9 Å, twist = −167.7°) (Fig. 5c and Extended Data Fig. 9).

### Reporting summary

Further information on research design is available in the Nature Portfolio Reporting Summary linked to this article.

## Data availability

The 'Ca. L. ossiferum' genome sequence (accession CP104013) was uploaded to GenBank, under BioProject ID PRJNA847409, BioSample accession SAMN28933922. Sub-tomogram averages (EMD-15987–EMD-15988), representative tomograms (EMD-15989–EMD-15993) and the corresponding tilt series (EMPIAR-11269) have been uploaded to the Electron Microscopy Data Bank and the Electron Microscopy Public Image Archive.

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

**Acknowledgements** We thank M. Mousavian, M. A. Mardini, K. Hager and A. Tramontano for technical assistance in the cultivation process and M. Melcher for assistance in the initial discovery of Lokiarchaea in shallow sediments; F. Sousa for advice on phylogenetic and evolutionary aspects; N. Kroll and J. Grünert for technical assistance in immuno-gold labelling and SEM; A. Oren for advice in naming the organism; J. Heebner and M. Swulius (Penn State College of Medicine) for sharing their Dragonfly segmentation protocol; the imaging platforms ScopeM at ETH Zürich and the Center for Microscopy and Image Analysis at the University of Zurich for instrument access; and J. Döhner and J. Hehl for advice on STED imaging. T.R.-O. thanks L. Pignaton for assistance with Extended Data Fig. 1. C.S. was supported by the European Research Council (AdG TACKLE, 695192) and the Austrian Science Fund (FWF): W1257. M.P. was supported by the European Research Council (CoG CryoET4Diversity, 101000232) and the NOMIS Foundation.

**Author contributions** T.R.-O. performed all cultivation-related experiments, as well as qPCR and RT–qPCR and FISH analyses. R.I.P.-T. assembled and analysed the genome and performed phylogenetic calculations and comparative genomic analyses. F.W. acquired, processed and analysed cryo-ET data, performed western blotting and immunofluorescence experiments. F.W. and J.X. developed the ribosome-identification approach. J.X. performed sub-tomogram averaging of the ribosome and developed the filament analysis workflow. A.K. performed SEM and TEM analyses. S.K.-M.R.R. advised on anaerobic cultivation. C.S. collected the original sediment cores and water samples. C.S. and M.P. were responsible for the conceptualization and supervision of the project. T.R.-O., F.W., R.I.P.-T., M.P. and C.S. wrote the manuscript with input from all of the authors.

**Competing interests** The authors declare no competing interests.

**Additional information**
**Correspondence and requests for materials** should be addressed to Martin Pilhofer or Christa Schleper.

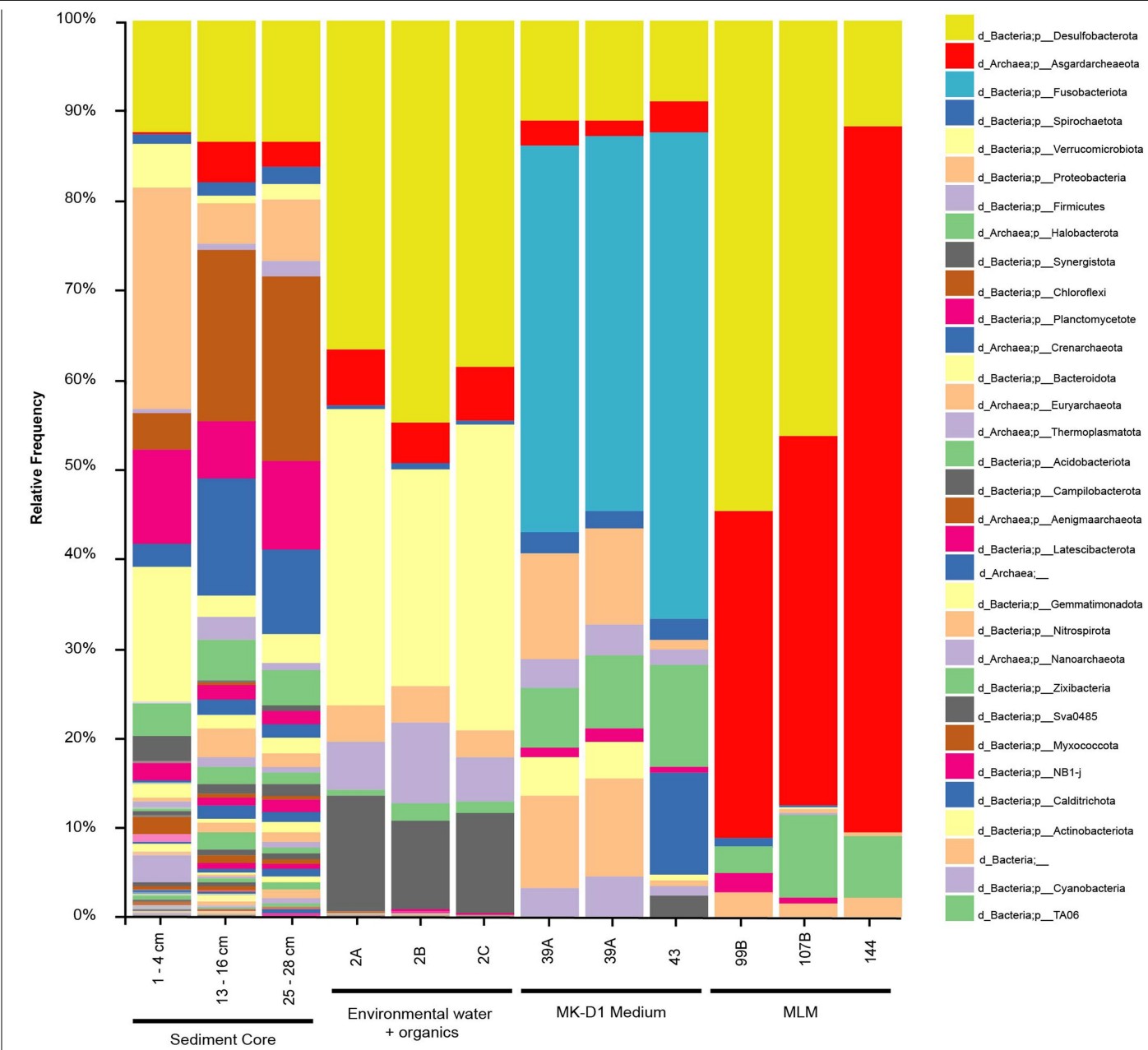

**Extended Data Fig. 1 | Microbial diversity profile of environmental samples (different brackish sediment core depths) and selected *L. ossiferum* enrichments.** Diversity estimations were made by amplicon sequencing using general prokaryotic 16S rRNA gene targeting primers (515f-806r). Taxonomic assignment was performed according to the SILVA database. Throughout the enrichment process, the community became more homogeneous, leading to higher lokiarchaeal relative abundances.

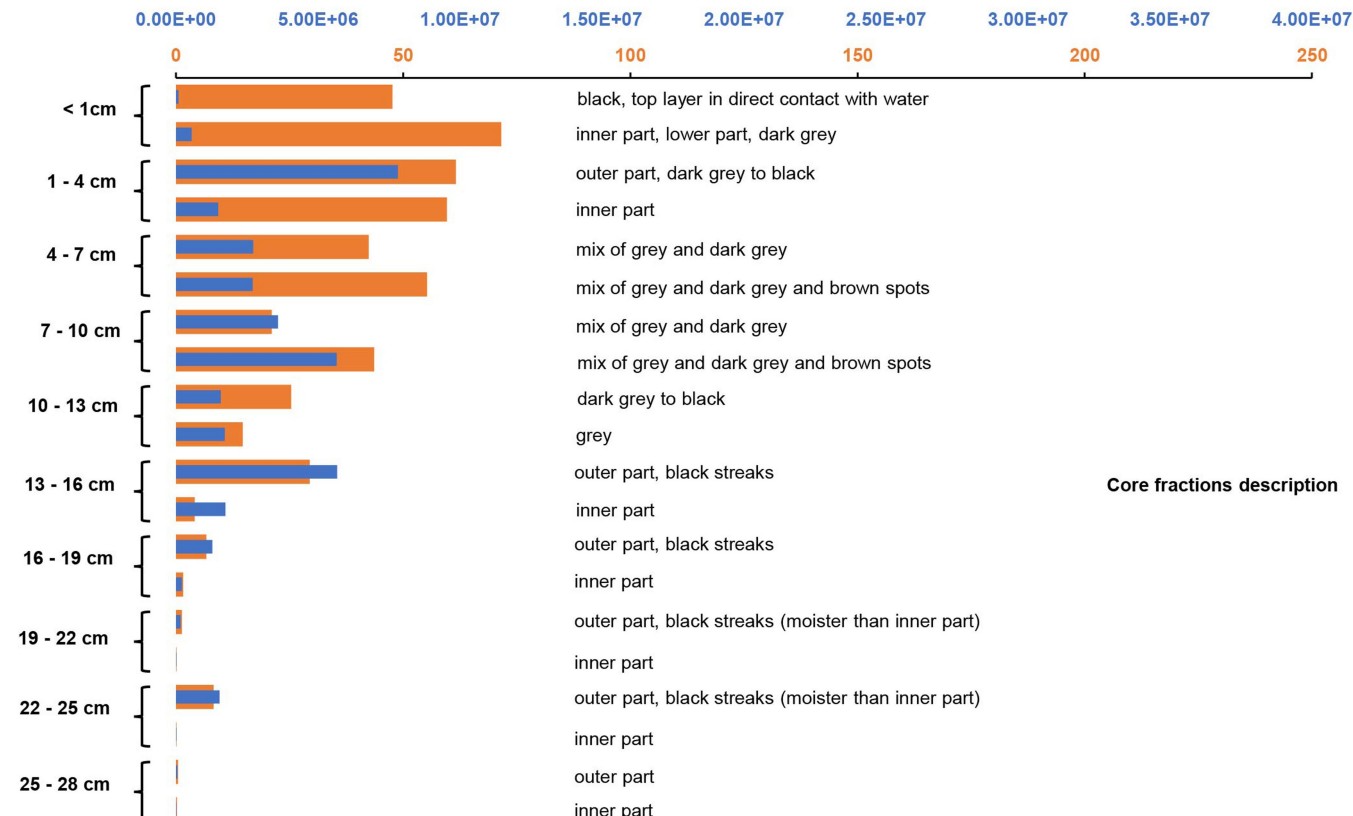

**Extended Data Fig. 2 | Lokiarchaea distribution at different depths in a brackish sediment core sampled in Piran, Slovenia.** Abundance was estimated by submitting DNA extracted from different depths to qPCR assays using primers targeting lokiarchaeal 16S rRNA genes. Criteria for selecting good candidate fractions for enrichments were the number of copies in relation to DNA content. The sample with the highest lokiarchaeal 16S rRNA gene copies/g ratio to DNA content was the 13–16 cm fraction, which was then selected as inoculum for cultivation.

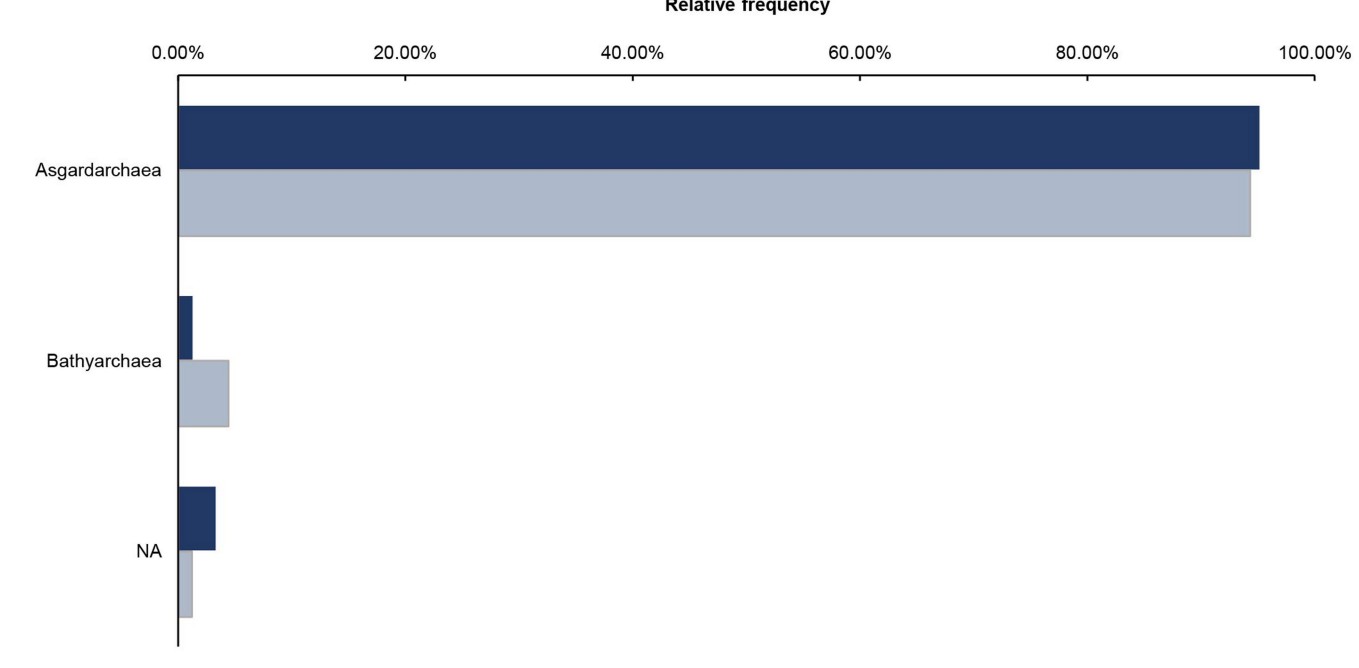

**Relative frequency**

■ Marine sediment DNA (3955 reads)    ■ Brackish sediment DNA (39419 reads)

**Extended Data Fig. 3 | Specificity of qPCR primers LkF and LkR to Asgard archaea 16S rRNA genes.** Amplicons were generated from environmental DNA and submitted to Illumina Miseq (300 PE) sequencing. SILVA was used for taxonomic classification. These analyses show that the LkF and LkR primers designed in this study are highly specific to Asgard archaea 16S rRNA genes.

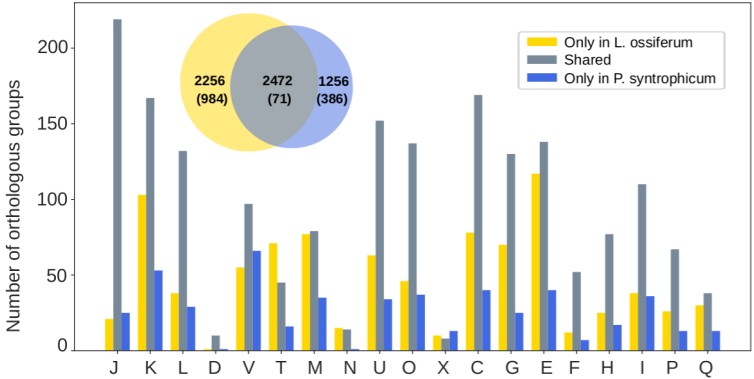

**Extended Data Fig. 4 | Functional annotation of groups of orthologous proteins encoded in *L. ossiferum* and '*Ca.* P. syntrophicum' genomes.** The venn diagram shows to scale the number of shared clusters of orthologous proteins between *L. ossiferum* and '*Ca.* P. syntrophicum' as well as genome-specific clusters. Numbers in parentheses indicate the number of clusters with no functional annotation. The graph shows that *L. ossiferum* is enriched in almost all categories compared to '*Ca.* P. syntrophicum', as expected for the larger genome size (>1.6 Mb difference). Functional categories: J, translation, ribosomal structure and biogenesis; K, transcription; L, replication, recombination and repair; D, cell cycle control, cell division and chromosome partitioning; V, defence mechanisms; T, signal transduction mechanisms; M, biogenesis of the cell wall, membrane or envelope; N, cell motility; U, intracellular trafficking, secretion and vesicular transport; O, posttranslational modification, protein turnover and chaperones; X, mobilome; C, energy production and conversion; G, carbohydrate transport and metabolism; E, amino acid transport and metabolism; F, nucleotide transport and metabolism; H, coenzyme transport and metabolism; I, lipid transport and metabolism; P, inorganic ion transport and metabolism; Q, secondary metabolites biosynthesis, transport and catabolism. For visualization purposes, clusters of orthologous proteins assigned to R (general function prediction only) and S (function unknown) categories are not shown. The numbers of clusters with only R and/or S assignment are: only in *L. ossiferum*, 405; shared, 617; only in '*Ca.* P. syntrophicum', 384.

Ca. Lokiarchaeum ossiferum

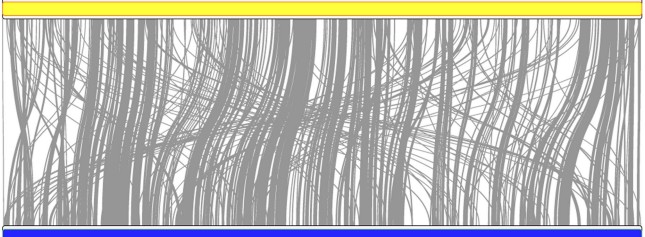

Ca. Prometheoarchaeum syntrophicum

**Extended Data Fig. 5 | Ideogram representation of syntenic regions (grey) between the genomes of *L. ossiferum* and '*Ca.* P. syntrophicum'.** Note that the genomes are represented in equal sizes (yellow and blue bars), although *L. ossiferum* has a 1.6 Mb larger genome.

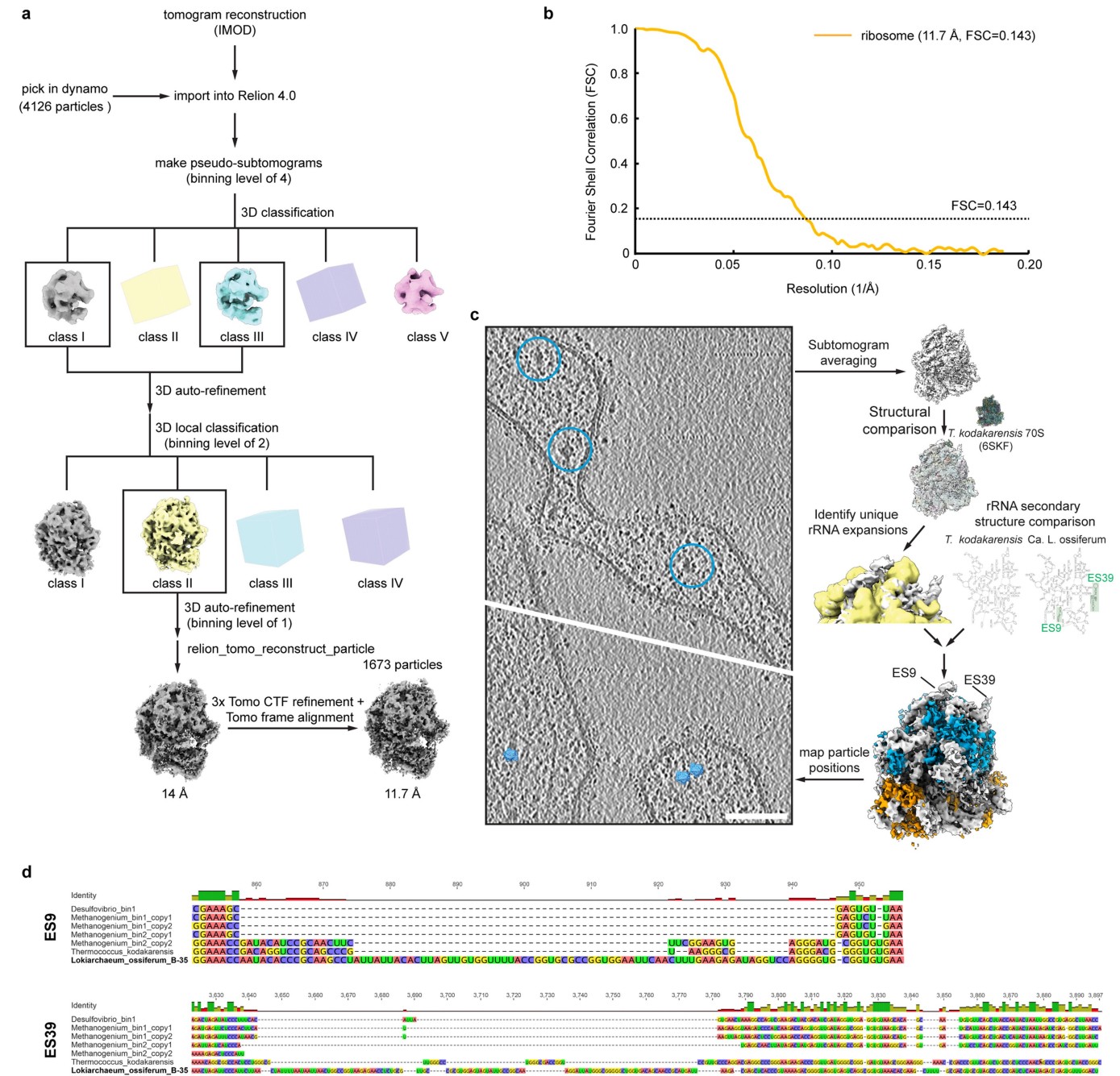

**Extended Data Fig. 6 | Sub-tomogram averaging of the ribosome and identification of rRNA expansion segments. a.** Flowchart for the sub-tomogram averaging strategy of the ribosome. For details see Methods. **b.** Gold-standard FSC curve of the sub-tomogram average. **c.** Scheme of the ribosome identification workflow. For details see Fig. 3 and Methods. **d.** Multiple sequence alignment of expansion segments (ESs) 9 and 39 of the LSU rRNA of *L. ossiferum* compared to sequences from *T. kodakarensis* and those present in the MAGs of the organisms in the enrichment culture (Desulfovibrio_bin1: 44 contigs, 100% complete, 0% contamination; Methanogenium_bin1: 56 contigs, 98.7% complete, 2.6% contamination; Methanogenium_bin2: 37 contigs, 98.7% complete, 1.9% contamination). The alignment reveals that ESs are only present in *L. ossiferum*.

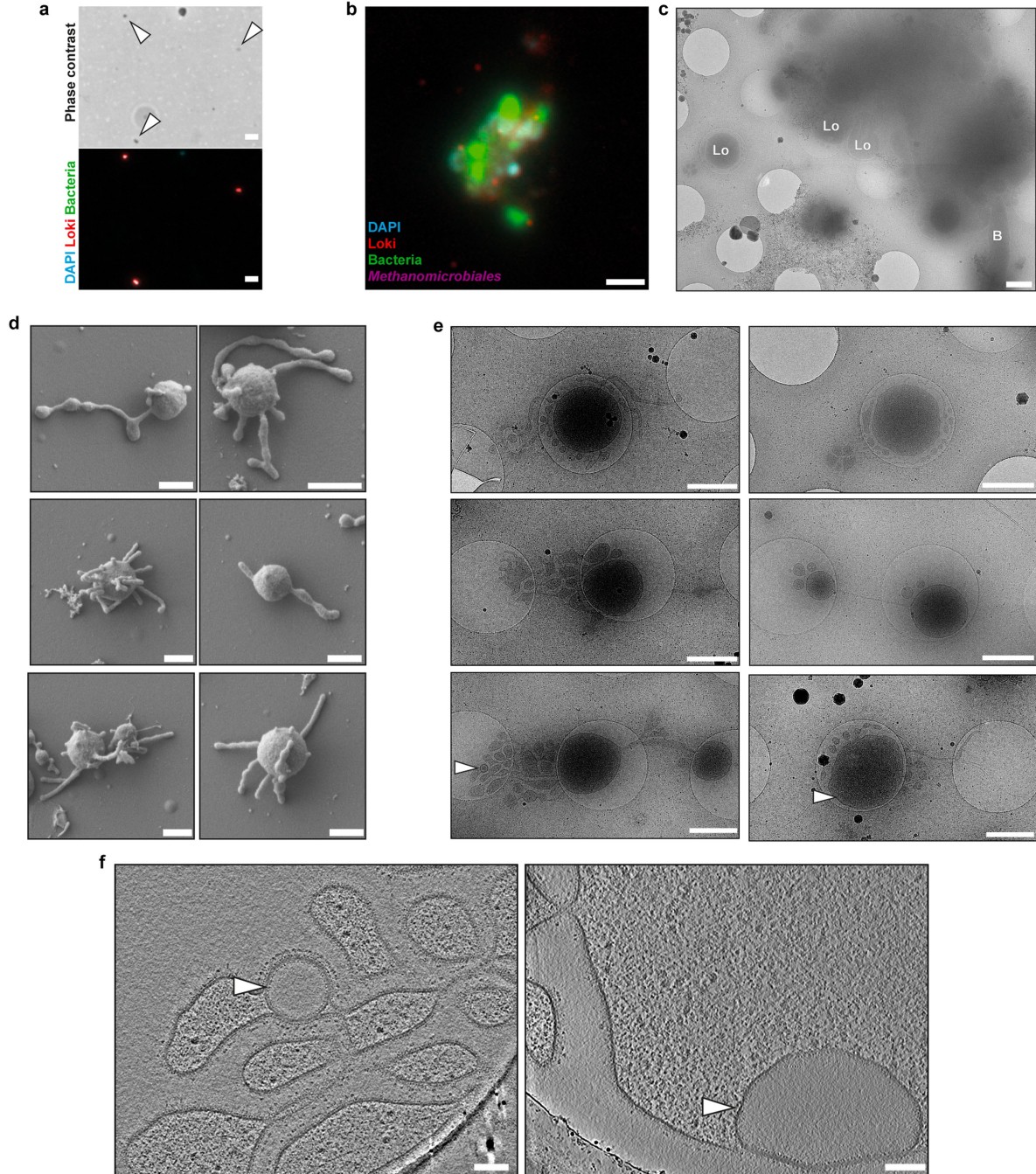

**Extended Data Fig. 7 | Potential cell-cell interactions, variability of cell shape and rare vesicles observed in cryo-tomograms. a**. FISH analysis of the enrichment cultures with DAPI (cyan) and nucleotide probes targeting Lokiarchaea (red) and bacteria (green) and the corresponding phase contrast image showing isolated *L. ossiferum* cells. The FISH experiments were performed five independent times, with similar results. Bar: 2 µm. **b/c**. Aggregates of cells were infrequently observed during imaging of the enrichment culture. FISH analysis (b) with DAPI (cyan) and nucleotide probes targeting Lokiarchaea (red), bacteria (green) and *Methanomicrobiales* (purple) shows potential interactions of cells from different species. A low-magnification 2D EM image (c) shows *L. ossiferum* cells (Lo) next to a bacterial cell (B).

Since the core of the cell assembly exceeded the thickness limitation of cryoET, no cryo-tomograms could be collected. Bars: 2 µm (b), 1 µm (c). **d/e**. Representative SEM (d) or low-magnification cryo-TEM images (e) showing different morphologies of *L. ossiferum* cells (n = 2 independent cultures). Both datasets revealed individual cell bodies that were always connected to at least one longer protrusion or vesicular structure. White arrowheads indicate the position of intracytoplasmic vesicles shown in f. Bars: 500 nm (d), 1 µm (e). **f**. Slices of cryo-tomograms (slice thickness 21.42 nm) showing the two intracytoplasmic vesicles observed in *L. ossiferum* protrusions (left) or a cell body (right), indicated by white arrowheads. Both vesicles seem fully enclosed by a membrane and show a low-density lumen. Bars: 100 nm.

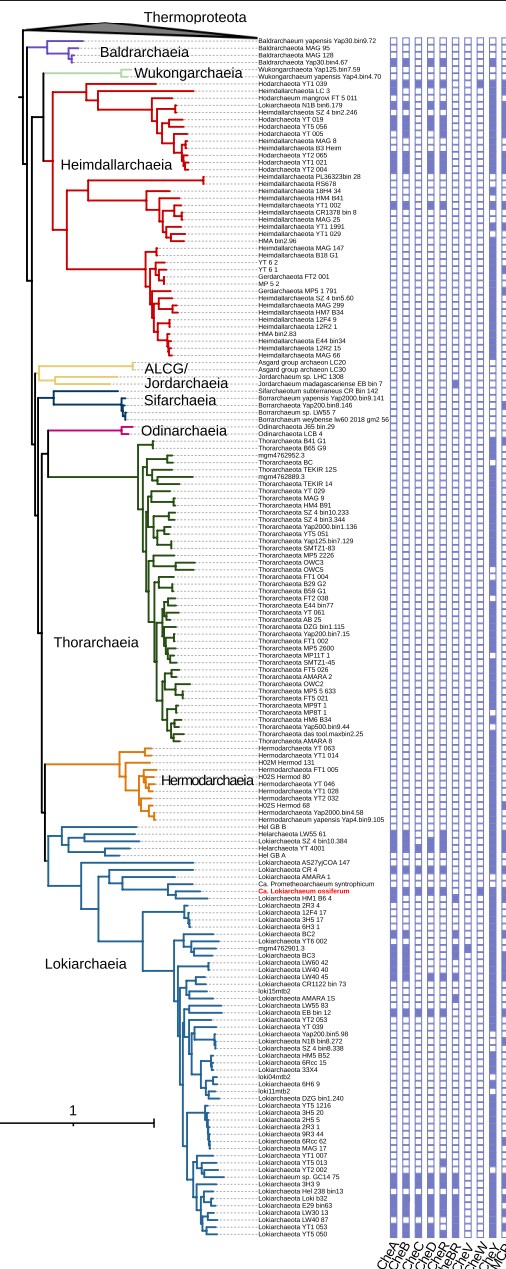

**Extended Data Fig. 8 | Presence/absence pattern of chemotaxis-related proteins in Asgardarchaeota genomes.** The chemotaxis-related proteins encoded in *L. ossiferum* (Supplementary Table 4) together with the chemotaxis protein CheV MBN1502358.1 and the chemotaxis accessory protein CheZ CK5345549.1 were used as queries in BLASTp searches (evalue: 1e$^{-10}$) against our Asgardarchaeota protein database followed by the annotation of putative chemotaxis proteins using BlastKOALA[87]. The presence/absence of proteins were mapped to the expanded phylogenetic tree shown in Fig. 2e. *L. ossiferum* is highlighted in red.

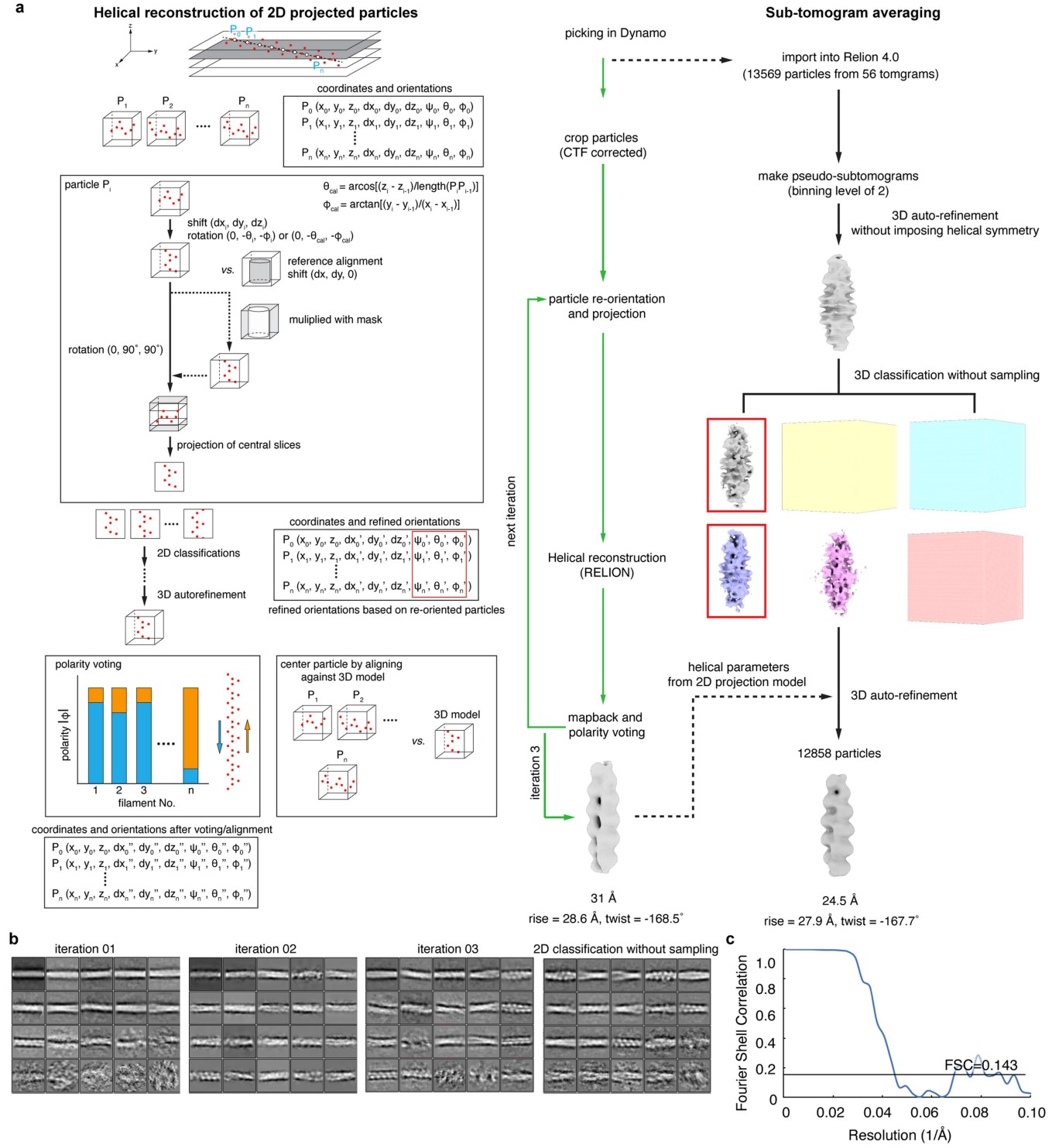

**a**

**Helical reconstruction of 2D projected particles**

coordinates and orientations

$P_0 (x_0, y_0, z_0, dx_0, dy_0, dz_0, \psi_0, \theta_0, \phi_0)$
$P_1 (x_1, y_1, z_1, dx_1, dy_1, dz_1, \psi_1, \theta_1, \phi_1)$
$P_n (x_n, y_n, z_n, dx_n, dy_n, dz_n, \psi_n, \theta_n, \phi_n)$

particle $P_i$

$\theta_{cal} = arcos[(z_i - z_{i-1})/length(P_iP_{i-1})]$
$\phi_{cal} = arctan[(y_i - y_{i-1})/(x_i - x_{i-1})]$

shift $(dx_i, dy_i, dz_i)$
rotation $(0, -\theta_i, -\phi_i)$ or $(0, -\theta_{cal}, -\phi_{cal})$

vs.    reference alignment
shift $(dx, dy, 0)$

muliplied with mask

rotation $(0, 90°, 90°)$

projection of central slices

2D classifications

coordinates and refined orientations

$P_0 (x_0, y_0, z_0, dx_0', dy_0', dz_0', \psi_0', \theta_0', \phi_0')$
$P_1 (x_1, y_1, z_1, dx_1', dy_1', dz_1', \psi_1', \theta_1', \phi_1')$
$P_n (x_n, y_n, z_n, dx_n', dy_n', dz_n', \psi_n', \theta_n', \phi_n')$

refined orientations based on re-oriented particles

3D autorefinement

polarity voting

center particle by aligning against 3D model

$P_1$    $P_2$    3D model
$P_n$    vs.

coordinates and orientations after voting/alignment

$P_0 (x_0, y_0, z_0, dx_0'', dy_0'', dz_0'', \psi_0'', \theta_0'', \phi_0'')$
$P_1 (x_1, y_1, z_1, dx_1'', dy_1'', dz_1'', \psi_1'', \theta_1'', \phi_1'')$
$P_n (x_n, y_n, z_n, dx_n'', dy_n'', dz_n'', \psi_n'', \theta_n'', \phi_n'')$

picking in Dynamo

crop particles
(CTF corrected)

particle re-orientation
and projection

next iteration

Helical reconstruction
(RELION)

mapback and
polarity voting

iteration 3

31 Å
rise = 28.6 Å, twist = -168.5°

**Sub-tomogram averaging**

import into Relion 4.0
(13569 particles from 56 tomgrams)

make pseudo-subtomograms
(binning level of 2)

3D auto-refinement
without imposing helical symmetry

3D classification without sampling

helical parameters
from 2D projection model

3D auto-refinement

12858 particles

24.5 Å
rise = 27.9 Å, twist = -167.7°

**b**

iteration 01    iteration 02    iteration 03    2D classification without sampling

**c**

FSC=0.143

**Extended Data Fig. 9 | Analysis of cytoskeletal filaments by helical reconstruction of 2D-projected particles and sub-tomogram averaging.** **a**. Flowchart of the *in situ* reconstruction of the cytoskeletal filament. For details see Methods. **b**. 2D classification of 2D-projected particles after different iterations of the reconstruction workflow and 2D classes obtained after the last iteration without sampling. **c**. Gold-standard FSC curve of the reconstruction after sub-tomogram averaging.

**a**

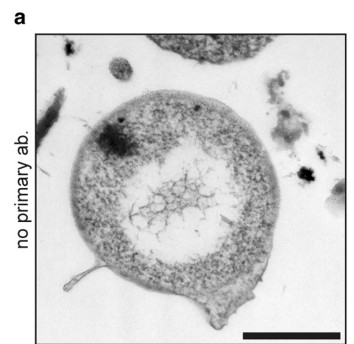

no primary ab.

**b**

α-Lokiactin

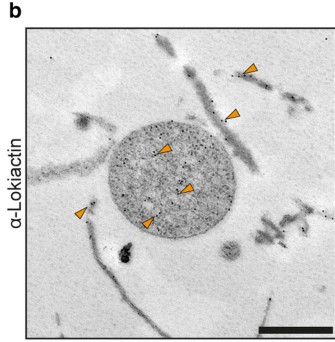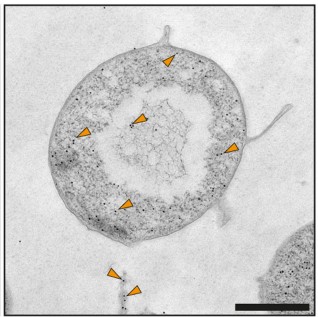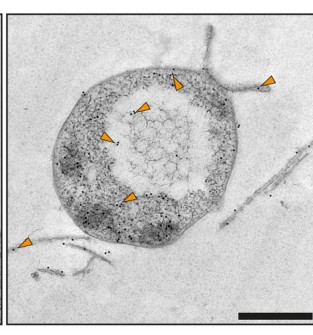

**Extended Data Fig. 10 | Immunogold localization of Lokiactin in**
***L. ossiferum*. a**. Conventional TEM micrograph of a *L. ossiferum* cell with no
anti-Lokiactin antibodies used. Experiments were performed with samples
derived from n = 2 independent cultures, with similar results. Bar: 500 nm.

**b**. Micrographs of different *L. ossiferum* cells immunogold labelled with a specific
anti-Lokiactin primary antibody (ab1, see Methods). Orange arrowheads point
out gold beads. Experiments were performed three independent times, with
similar results (n = 3). Bars: 500 nm. See also Supplementary Table 10.

Martin Pilhofer

# Reporting Summary

## Statistics

For all statistical analyses, confirm that the following items are present in the figure legend, table legend, main text, or Methods section.

| n/a | Confirmed | |
|---|---|---|
| ☐ | ☒ | The exact sample size (*n*) for each experimental group/condition, given as a discrete number and unit of measurement |
| ☐ | ☒ | A statement on whether measurements were taken from distinct samples or whether the same sample was measured repeatedly |
| ☒ | ☐ | The statistical test(s) used AND whether they are one- or two-sided *Only common tests should be described solely by name; describe more complex techniques in the Methods section.* |
| ☒ | ☐ | A description of all covariates tested |
| ☒ | ☐ | A description of any assumptions or corrections, such as tests of normality and adjustment for multiple comparisons |
| ☐ | ☒ | A full description of the statistical parameters including central tendency (e.g. means) or other basic estimates (e.g. regression coefficient) AND variation (e.g. standard deviation) or associated estimates of uncertainty (e.g. confidence intervals) |
| ☒ | ☐ | For null hypothesis testing, the test statistic (e.g. *F*, *t*, *r*) with confidence intervals, effect sizes, degrees of freedom and *P* value noted *Give P values as exact values whenever suitable.* |
| ☒ | ☐ | For Bayesian analysis, information on the choice of priors and Markov chain Monte Carlo settings |
| ☒ | ☐ | For hierarchical and complex designs, identification of the appropriate level for tests and full reporting of outcomes |
| ☒ | ☐ | Estimates of effect sizes (e.g. Cohen's *d*, Pearson's *r*), indicating how they were calculated |

*Our web collection on statistics for biologists contains articles on many of the points above.*

## Software and code

Policy information about availability of computer code

| | |
|---|---|
| Data collection | CFX Maestro version 2.3 software for qPCR data collection; GRYPHAX version 1.1.8.153 for FISH imaging; SerialEM 4.0 for Cryo-ET data collection, Zeiss ZenBlue 3.5 for Airyscan data collection; Leica LASX for STED data collection |
| Data analysis | Fiji/ImageJ (v2.3.0/1.53q) for light microscopy analysis; IMOD (v4.11) for tomogram reconstruction and visualization; Dragonfly (v2022) for tomogram segmentation; RELION-4.0 for ribosome sub-tomogram averaging and actin reconstruction; ChimeraX (v1.3) for visualization; Dynamo for particle picking; R2DT for rRNA secondary structure prediction; ZenBlue 3.5 for Airyscan image processing; Huygens Professional (v22.04) for STED deconvolution; QIIME2 for amplicon analysis; Trimmomatic v.0.36 was used for trimming; Short reads were assembled with SPAdes v3.15.2; Binsanity, MaxBin2, MetaBAT, CONCOCT were used for genome binning; DAS tool for bin dereplication; Bonito 0.3.6 for nanopore reads basecalling; Porechop v0.2.4 and NanoFilt v2.8.0 for adapter removal and quality control of long reads; Flye v2.8.3-b1695 for assembly of long reads; Pilon for genome polishing; MAFFT v7.427 for sequence alignment; trimAl for alignment trimming; IQ-TREE 2.0 for phylogenetic tree reconstruction; BMGE for sequence alignment. |

For manuscripts utilizing custom algorithms or software that are central to the research but not yet described in published literature, software must be made available to editors and reviewers. We strongly encourage code deposition in a community repository (e.g. GitHub). See the Nature Portfolio guidelines for submitting code & software for further information.

## Data

Policy information about availability of data

All manuscripts must include a data availability statement. This statement should provide the following information, where applicable:
- Accession codes, unique identifiers, or web links for publicly available datasets
- A description of any restrictions on data availability
- For clinical datasets or third party data, please ensure that the statement adheres to our policy

> The accession number of the genome is CP104013 (GenBank), under BioProject ID: PRJNA847409, BioSample accession: SAMN28933922.
> Sub-tomogram averages (accession codes: EMD-15987 - EMD15988), example tomograms (accession codes: EMD-15989 - EMD-15993) and corresponding tilt-series (accession code: EMPIAR-11269) have been uploaded to the Electron Microscopy Databank or the Electron Microscopy Public Image Archive.
> Other datasets used in this study from the Electron Microscopy Databank: EMD-13448, EMD-11976
> Other datasets used in this study from the Protein Data Bank (PDB): 6SKF, 3J8A, 5MW1

## Human research participants

Policy information about studies involving human research participants and Sex and Gender in Research.

| Reporting on sex and gender | N/A |
|---|---|
| Population characteristics | N/A |
| Recruitment | N/A |
| Ethics oversight | N/A |

Note that full information on the approval of the study protocol must also be provided in the manuscript.

# Field-specific reporting

Please select the one below that is the best fit for your research. If you are not sure, read the appropriate sections before making your selection.

☒ Life sciences          ☐ Behavioural & social sciences          ☐ Ecological, evolutionary & environmental sciences

For a reference copy of the document with all sections, see nature.com/documents/nr-reporting-summary-flat.pdf

# Life sciences study design

All studies must disclose on these points even when the disclosure is negative.

| Sample size | No sample size calculations were performed. Imaging experiments were performed on samples derived from at least 2 independent cultures (as stated in the figure legends). For sub-tomogram averaging, please see ED Fig. 6 and 9 and the methods section for the number of particles and particle selection. |
|---|---|
| Data exclusions | Cultures were selected based on detectable exponential growth for experiments. No data were excluded from the analysis. For exclusion of particles from sub-tomogram averaging, please see ED Fig. 6 and 9 and the methods section. |
| Replication | FISH, RT-qPCR, SEM, TEM, IF and cryoET and immunogold localization were all performed from two to five independent times (as stated in the figure legends), with all attempts at replication being succesful. |
| Randomization | Randomization is not relevant for the current study as it does not involve participant groups. |
| Blinding | Blinding is not relevant to the present study, as it is cultivation based and the researchers involved need to to verify samples and controls for each experiment. |

# Reporting for specific materials, systems and methods

We require information from authors about some types of materials, experimental systems and methods used in many studies. Here, indicate whether each material, system or method listed is relevant to your study. If you are not sure if a list item applies to your research, read the appropriate section before selecting a response.

## Materials & experimental systems

| n/a | Involved in the study |
|---|---|
| ☐ | ☒ Antibodies |
| ☒ | ☐ Eukaryotic cell lines |
| ☒ | ☐ Palaeontology and archaeology |
| ☐ | ☒ Animals and other organisms |
| ☒ | ☐ Clinical data |
| ☒ | ☐ Dual use research of concern |

## Methods

| n/a | Involved in the study |
|---|---|
| ☒ | ☐ ChIP-seq |
| ☒ | ☐ Flow cytometry |
| ☒ | ☐ MRI-based neuroimaging |

## Antibodies

| | |
|---|---|
| Antibodies used | Custom antibodies were obtained using the services of Eurogentec (order number 1000997858). They were produced in rabbits (ab1: CTFYTDLRVDPSEHPV - lot number 2110517; ab2: CSKNGFAGEDQPRSVF - lot number 2110516) and used at dilutions of either 1:100/1:500 (for IF) or 1:1000 (for WB).<br>Secondary antibodies for immunofluorescence and Western blotting experiments were: donkey anti-rabbit AF647, Invitrogen A-31573 (1:500 diluted), goat anti-rabbit abberior STAR 580, abberior ST580-1002 (1:200 diluted) and goat anti-rabbit HRP, Invitrogen 31460 (1:5000 diluted). |
| Validation | The identity of the peptides was confirmed by LC-MS analysis. The antibodies were validated through ELISA assays using the services of Eurogentec and additionally tested on L. ossiferum cell lysate by Western blotting (see Fig. 5f). |

## Animals and other research organisms

Policy information about studies involving animals; ARRIVE guidelines recommended for reporting animal research, and Sex and Gender in Research

| | |
|---|---|
| Laboratory animals | *For laboratory animals, report species, strain and age OR state that the study did not involve laboratory animals.* |
| Wild animals | *Provide details on animals observed in or captured in the field; report species and age where possible. Describe how animals were caught and transported and what happened to captive animals after the study (if killed, explain why and describe method; if released, say where and when) OR state that the study did not involve wild animals.* |
| Reporting on sex | *Indicate if findings apply to only one sex; describe whether sex was considered in study design, methods used for assigning sex. Provide data disaggregated for sex where this information has been collected in the source data as appropriate; provide overall numbers in this Reporting Summary. Please state if this information has not been collected. Report sex-based analyses where performed, justify reasons for lack of sex-based analysis.* |
| Field-collected samples | Collection of the sediment samples and the enrichment culture are described in the Methods section. |
| Ethics oversight | *Identify the organization(s) that approved or provided guidance on the study protocol, OR state that no ethical approval or guidance was required and explain why not.* |

Note that full information on the approval of the study protocol must also be provided in the manuscript.

