## [Peer Review File · Nature]

Manuscript Title: Actin cytoskeleton and complex cell architecture in an Asgard archaeon

Reviewer Comments & Author Rebuttals

Reviewer Reports on the Initial Version:

Referees' comments:

Referee #1 (Remarks to the Author):

This paper reports the first ultrastructural analysis of an Asgard archaeon. After identifying a suitable sampling site rich in Asgard archaeal DNA, the authors manage to establish an enrichment culture and identify optimized growth conditions that allows them to enrich a single Asgard archaeon to approx. 80% relative abundance. Genome sequencing and analysis suggests that this archaeon, named *Candidatus Lokiarchaeum ossiferum*, is distinct from a previously enriched Asgard archaeal species. After providing first insight into the nature of *L. ossiferum* cells by FISH, the authors set out to obtain high-resolution structural information by cryo-EM tomography. They use ultrastructural features to distinguish *L. ossiferum* cells from other archaeal and bacterial cells in the enrichment culture. In particular, they reconstruct the structure of *L. ossiferum* ribosomes by sub-tomogram averaging and verify that they exhibit additional densities corresponding to the typical supersized eukaryotic-like expansion segments of the Asgard archaeal rRNA. A detailed analysis of the cryo-EM tomograms (and SEM images) reveals that *L. ossiferum* exhibits highly variable cell morphology, characterized by long and often branching protrusions. The cells contain various electron-dense assemblies. In particular, they display long filamentous structures that are interpreted as lokiactin polymers, based on the similarity of their in-situ cryo-EM structure with that of F actin and the detection of lokiactin-specific signals by Western blotting, immunogold labeling and immunofluorescence imaging.

Overall, this is an interesting paper that for the first time provides insight into the cellular architecture of a representative of the Asgard archaea, the prokaryotic lineage most closely related to eukaryotes. The data suggest that *L. ossiferum* possesses an actin network that may be involved in regulating the complex cell shape of this species. The data are largely of high quality and presented clearly in the manuscript. The findings are novel and interesting to a broad audience, but there are several issues that should be addressed before publication.

1. How sure is it that the complex, pleomorphic cell shapes observed in fact represent that native state of *L. ossiferum* and are not an artifact of the culture conditions? Can similar types of cells be observed in the sediment samples used as an inoculum of the enrichment culture? What is the ratio of highly branched/complex cells versus largely round-shaped cells? It would be interesting to see a phase contrast/DIC image of a larger field of cells or a quantification of the morphologies observed, even though a precise categorization is probably difficult to achieve.
2. It would be helpful to the reader to see a phase contrast/DIC image of the field shown in Fig. 3a.

3. The authors state that the filaments observed are often close the membrane (Line 290). However, in the large cell shown in Fig. 4d, they appear to largely located distal to the membranes. Is there a way to quantify the distribution of the filaments within the cell more precisely? Would it be possible to provide movies with 3D rendered tomograms to better visualize the filament network?

4. The evidence supporting the conclusion that the filaments observed represent lokiactin is not convincing, especially since *L. ossiferum* encodes several different actin homologs. The resolution of the in-situ structure obtained is not sufficiently high to clearly identify the protein as lokiactin, and the fact that the production of lokiactin in *L. ossiferum* can be shown by Western blotting and immunolabeling approaches does not prove that the protein in fact forms the large filaments detected.

The authors should provide additional evidence or tone down their conclusions.

5. The cell shown as a negative control for the immunogold labeling experiment is very dark, so that it would be difficult to see non-specific binding of the gold beads. A better control and potentially some kind of quantification is required.

6. The resolution of the immunofluorescence images in Fig. 5g is relatively low, so that it is difficult to recognize distinct structures. The authors should use super-resolution fluorescence imaging to better resolve the structures and thus provide further evidence of their filamentous nature.

Other comments/suggestions:

- Line 118: "encoding 5,119 proteins"
- Line 121: "and even three ribosomal RNA operons"
- Lines 126/136: What are "unique orthologous proteins"? What are they orthologous to?
- Fig. S11: It would be helpful to see the position of eukaryotic actins in the this tree.

Referee #2 (Remarks to the Author):

The manuscript by Rodrigues-Oliveira et al. reports the enrichment, genome, and ultrastructural characterization of *Ca. Lokiarchaeum ossiferum*, the second ever member of Asgard archaea, the closest relatives of Eukaryotes.

This is per se a giant achievement, as we know next to nothing about these intriguing microorganisms.

Moreover, the authors make it very clear how their model offers undeniable advantages over *Prometheoarchaeum* that was isolated in coculture last summer. It thrives in easy to access sites (shallow water sediment in Slovenia), and the authors successfully found optimal growth conditions (7-14 days generation time and high cell densities).

But most importantly, the authors take on an incredible technical challenge with imaging the cell ultrastructure of this uncultured archaeon, providing unprecedented insights into its organization.

The quantity and quality of the images is stunning. Also, the authors do not stop there but complement these images with many other techniques, making this work extremely rich and exhaustive, at the limit that can be done on an uncultured prokaryote. I particularly liked the solution to identify *L. ossiferum* with respect to other archaea and bacteria present in the enrichment by linking it to a specific structure of the ribosome, a very nice idea that opens the way to identify Asgard archaea in their natural environment, as FISH cannot be combined with high resolution imaging.

So, overall, I congratulate the authors for this very nice work which we'll be of great impact to the scientific community.

I have just a few questions/comments:

First, two things that the authors may want to elaborate just a bit further in the text:

- How many archaea have been characterized structurally in such depth and how specific are these protrusions and cell complexity in asgard archaea?
- same with the ribosomal expansion, how do you know that they are not present in yet to identify archaea, and in that case how can you be sure to link if to asgards in the environment?

Line 117: you may want to state here that you obtained a closed genome in one contig, although this is already in the figure it might be good to say it also here.

Line 125: the genome is significantly larger than Promethe, which is very close phylogenetically. You may want to reassure further the readers that you did not make any mistake of assembly here. Did you check that they do not have a double of ribosomal proteins for example? Also, it may be informative to indicate the genome sizes of asgard members on Figure 2e, or even only the lokiarchaeia. This can also be briefly stated in the text. Did you sequence the genomes of the other two members (halodesulfo and methanogenium) enrichment other than only the 16S?

Line 169: refer to figure 3c already at the end of this sentence.

Line 173: the authors may want to clarify once more the technical challenges that prevented a correlated FISH-cryoET approach.

Line 266: the Etymology before the conclusions looks a bit bizarre, I would move it further down.

Line 281: please nuance a bit the statement that the actin-based cytoskeleton has never been visualized. Or explain better here your advances with respect to the Prometheoarchaeum paper.

Line 298: see my second comment and maybe nuance a bit here or develop further.

Referee #3 (Remarks to the Author):

Review: Rodrigues-Oliveira et al., "Complex cytoskeleton and cell architecture in an Asgard

archaeon", Nature, manuscript: 427877 (2022)

The authors report the enrichment of a new Asgard archaeon, which they name Lokiarchaeum ossiferum. They sequence the complete genome and report a number of interesting statistic and findings derived from the sequence. The authors then progress to imaging and use electron cryo-tomography and some FISH and immunofluorescence.

One difficulty they overcome very elegantly is that the culture contained a number of organisms and they needed to be sure which cells are which in their images. They used "visual proteomics" to achieve this, which, to my knowledge, is the first time the method has been used in this way. They determined the (low resolution) sub-tomogram-averaged structure of the ribosome from the cells they thoughts were L. ossiferum cells and determined that the structure they obtained showed the known expansion segments 9 & 39 in the RNA that are characteristic of Asgard organisms. Most importantly, the resulting tomograms also show extensive and beautiful cellular protrusions on the surface of the cells. Because these are thin, the images are very convincing and it is also possible to discern chemotaxis arrays, ribosomes and filaments in these protrusions and in small cell bodies. They then go on to determine a low-resolution structure of the filaments and produce a map that is consistent with the filaments being F-actin as formed by "lokiactin".

It is important for me to point out two things: first, this work is difficult and only one (low quality) study has previously reported an enrichment culture of an Asgard organism. No previous studies have provided images that show with such clarity and certainty that Asgard organisms can have surface protrusions and that these contain F-actin-like filaments. Second, the protrusions are supportive of a theory of how eukaryotes came about, the now-famous inside-out theory by Baum & Baum (ref 50). It is clear that we have to think about this theory much harder now and it is also clear that experimental progress will now be rapid, not least based on the study under review here.

I review many papers, but I can state that this is the most exciting manuscript I had the pleasure of being asked to review for a number of years: it is important, well-performed, beautiful to look at and easy to follow and understand. The manuscript is well written and presented. The two corresponding authors are experts in their fields (archaeal microbiology and cellular tomography) and the work is of very high quality. While I am very supportive of this getting published asap, below are a few comments and questions, of which some might seem overly critical, but I wanted to be extra helpful. Please publish.

Some comments and questions in no particular order:

1: (title): I am not sure the word "complex" is justified (yet) for the cytoskeleton imaged in the study. A suggestion: "Enriched Asgard archaea reveal cell surface protrusions that contain actin filaments"

36: How are the filaments shown here principally different or more "complex" than, let's say ParM actin-like filaments that cross the body of E. coli cells? (related to previous point)

53: ESCRT-III proteins have just been identified in bacteria ... (PspA, Vipp1 ..)

54: Intro could mention that bona fide F-actin has been found outside Asgard (crenactin).

65: To my knowledge this model was first mentioned in ref 50 (Baum & Baum) and needs to be mentioned/cited here and not at the very end of the manuscript.

65: Why would that theory need ESPs? Could also have been done with proteins whose genes were lost afterwards. Not a good argument.

93: Those enrichments of 25-80% are impressive – have single cell inoculations been tried after cells were recognised? And also with the putative syntrophic partners?

108: Unit of 2.5? Unit needs to be mentioned first (minor issue, sorry).

162: Possible that the fragility of the cells causes them to fragment on the grids? How can we be sure that the cells really look like this when not blotted to near-dryness on EM grids? Superresolution light microscopy with a membrane dye? Cryo slice-and-view (black face imaging)? This is something that could be done quite quickly and I think could add additional certainty.

173: What were those challenges that prevented FISH-CLEM? Would have been nice and powerful ...

177: How many particles were averaged from how many cells? (Methods say 4,126 particles, and this should be mentioned in the main text). Also, it seems from the methods that no ribosomes were reconstructed from the non-L. ossiferum cells? Is that not a (theoretical) weakness (lack of control)?

191: I am not sure what the scanning EM adds here. Sample preparation is harsh. Volume EM (cryo slice-and-view/blockface imaging) would have been better ...

204: Filament branching is exciting! Any signs of branching proteins in the genome?

212: Surface structures: where L. ossiferum cells contact other cells, any sign that these structures are involved?

227: I think it would only be justified to suggest that the filaments scaffold the protrusions if filaments were found in most, if not all membrane structures. More careful wording I think is needed here. It is quite possible that the protrusions are caused by some other proteins and the filaments just happen to be inside to do something else, for example.

251: A much more involved way to make sure the raised antibodies do what they are supposed to do would have been to make the various actins recombinantly, and to see how they bind. Not suggesting that this needs to be done, just a suggestion.

254: And FISH-CLEM again was not possible? Why?

258-264: Not sure this is needed. It is speculative and the existence of gelsolin and profilin homologues could just be mentioned here.

266: *L. ossiferum*: don't the authors suggest that most likely all Asgard organisms have actin filaments? Does the new name make sense in light of this since it will probably not be a distinguishing feature?

280: I object to the assertion that the first actin-cytoskeleton has been discovered, if that was meant. Of course, actins are present throughout the bacteria and also archaea, and crenactin forms F-actin filaments that are structurally indistinguishable from eukaryotic F-actin. Please re-phrase (I know the second sentence is meant to be conditioned by the first, but this is not made very clear I think).

291-293: Again, I am not sure I would necessarily take away from the pictures that *L. ossiferum* actin shapes membranes and is a scaffold (for example Figure 4j). More careful wording is probably safer right now.

304: The authors mention earlier that they think it is unlikely that cell-cell contacts are needed for syntrophy. That weakens the statement here, I think. And, why are no cell contacts shown by cryo-ET? Too few?

307: The Baum & Baum model (ref 50) could be described in more detail since the data shown here is very supportive of what that theory predicted (even the picture!).

310: How would compartmentalisation work? The protrusions would need to be sealed off, no? Any evidence for that?

355: Data availability: will raw images that were used to reconstruct tomograms be deposited in EMPIAR? This would be really helpful, thank you!

619-: Wouldn't it have been better to do the structure of the filaments in Relion 4 directly from the pseudo sub-tomograms (no helical symmetry or only later). The projection analysis used will severely degrade resolution as all the errors that went in during initial tomogram reconstruction will persist, if I understood the method correctly.

Figure 5d: Since crenactin has the same structure as F-actin, I think it is slightly unhelpful to make it an outgroup. F-actin exists outside Asgard. Bacterial actins and other actins may need to be included to not give the wrong impression. Figure S11 attempts this, but again it only looks at sequences and disregards structural conservation, which is more important for function and evolution, since sequences diverge more quickly than structures.

General question: would you dare speculate why the protrusions are there? What is their function? They are not there to make eukaryotes, surely ...

Figure S10: no protrusions visible here? And, as is often the case, the immunogold labelling does not add much certainty ...

Author Rebuttals to Initial Comments:

Referees' comments:

Referee #1 (Remarks to the Author):

This paper reports the first ultrastructural analysis of an Asgard archaeon. After identifying a suitable sampling site rich in Asgard archaeal DNA, the authors manage to establish an enrichment culture and identify optimized growth conditions that allows them to enrich a single Asgard archaeon to approx. 80% relative abundance. Genome sequencing and analysis suggests that this archaeon, named *Candidatus Lokiarchaeum ossiferum*, is distinct from a previously enriched Asgard archaeal species. After providing first insight into the nature of *L. ossiferum* cells by FISH, the authors set out to obtain high-resolution structural information by cryo-EM tomography. They use ultrastructural features to distinguish *L. ossiferum* cells from other archaeal and bacterial cells in the enrichment culture. In particular, they reconstruct the structure of *L. ossiferum* ribosomes by sub-tomogram averaging and verify that they exhibit additional densities corresponding to the typical supersized eukaryotic-like expansion segments of the Asgard archaeal rRNA. A detailed analysis of the cryo-EM tomograms (and SEM images) reveals that *L. ossiferum* exhibits highly variable cell morphology, characterized by long and often branching protrusions. The cells contain various electron-dense assemblies. In particular, they display long filamentous structures that are interpreted as lokiactin polymers, based on the similarity of their in-situ cryo-EM structure with that of F actin and the detection of lokiactin-specific signals by Western blotting, immunogold labeling and immunofluorescence imaging.

Overall, this is an interesting paper that for the first time provides insight into the cellular architecture of a representative of the Asgard archaea, the prokaryotic lineage most closely related to eukaryotes. The data suggest that *L. ossiferum* possesses an actin network that may be involved in regulating the complex cell shape of this species. The data are largely of high quality and presented clearly in the manuscript. The findings are novel and interesting to a broad audience, but there are several issues that should be addressed before publication.

R: Thank you for this positive feedback!

1. How sure is it that the complex, pleomorphic cell shapes observed in fact represent that native state of *L. ossiferum* and are not an artifact of the culture conditions? Can similar types of cells be observed in the sediment samples used as an inoculum of the enrichment culture? What is the ratio of highly branched/complex cells versus largely round-shaped cells? It would be interesting to see a phase contrast/DIC image of a larger field of cells or a quantification of the morphologies observed, even though a precise categorization is probably difficult to achieve.

R: This is an interesting comment. It is indeed possible that the cell shape of *L. ossiferum* could be different depending on environmental or growth conditions. However, observing the cells in the original sediment is currently impossible due to the abundant presence of sediment particles as well as the low relative and absolute abundances of *Lokiarchaea* (at least when compared to the enrichment cultures). Additionally, while we cannot say that the cell shapes detected are the only ones that *L. ossiferum* adopts, these are the ones detected in our enrichment process. We agree that further investigations will be needed to study the variability of cell shapes in *Lokiarchaea* under different conditions. In this sense, every microbe characterized by cultivation suffers from a possible bias when it comes to determining its cell shape.

To address the reviewer's request, we now provide additional representative SEM and cryoTEM examples in Extended Data Fig. 7d/e to illustrate the pleomorphic cell shape. Both datasets revealed individual cell bodies always connected to at least one protrusion (stated in figure legend). Please also see the response to Reviewer 3.

2. It would be helpful to the reader to see a phase contrast/DIC image of the field shown in Fig. 3a.

R: We have included now the requested figure, by showing a FISH image and its corresponding phase contrast in Extended Data Fig. 7a. Please note that the harsh permeabilization required for FISH in combination with the high fragility of the cell makes it impossible to assess cell shape in these samples (see also our responses to Reviewers 2 and 3 regarding FISH/cryoET).

3. The authors state that the filaments observed are often close the membrane (Line 290). However, in the large cell shown in Fig. 4d, they appear to largely located distal to the membranes. Is there a way to quantify the distribution of the filaments within the cell more precisely? Would it be possible to provide movies with 3D rendered tomograms to better visualize the filament network?

R: We have now added a segmentation of the cell body shown in Fig. 4c/d to Supplementary movie 1. Unfortunately, the thickness of most cell bodies (and the high density of the cytoplasm that also contains e.g. DNA) results in comparatively noisy data, which currently precludes us from quantitatively assessing filament distribution by cryoET. In immuno-fluorescence experiments, we typically observe staining in the periphery of the cell but cannot assess the precise distance to the membrane. We have therefore changed the wording in line 282 to: "often in the periphery".

4. The evidence supporting the conclusion that the filaments observed represent lokiactin is not convincing, especially since *L. ossiferum* encodes several different actin homologs. The resolution of the in-situ structure obtained is not sufficiently high to clearly identify the protein as lokiactin, and the fact that the production of lokiactin in *L. ossiferum* can be shown by Western blotting and immunolabeling approaches does not prove that the protein in fact forms the large filaments detected. The authors should provide additional evidence or tone down their conclusions.

R: We now provide additional evidence by showing that the Lokiactin is by far the most highly expressed gene among all four homologs found in the genome. We demonstrate this by RT-qPCR analysis in two independent cultures and have included the new data as Fig. 5e. We also improved the sub-tomogram average (Fig. 5c, Extended Data Fig. 9), which clearly shows an F-actin like filament (see also the response to Reviewer 3).

We also would like to emphasize that the peptide antibodies were designed against two unique regions of the Lokiactin protein. Finally, and importantly, we now provide STED images that show clear filamentous assemblies labeled by the Lokiactin antibody (see response No. 6 below).

5. The cell shown as a negative control for the immunogold labeling experiment is very dark, so that it would be difficult to see non-specific binding of the gold beads. A better control and potentially some kind of quantification is required.

R: We agree and we have exchanged the control for the immunogold labeling to an image where non-specific binding would be better observed. Furthermore, we have added a quantification (which includes additional negative controls) to Suppl. Table 10.

6. The resolution of the immunofluorescence images in Fig. 5g is relatively low, so that it is difficult to recognize distinct structures. The authors should use super-resolution fluorescence imaging to better resolve the structures and thus provide further evidence of their filamentous nature.

R: Please note that these experiments detect single actin filaments by indirect immunofluorescence (i.e. using a primary and a secondary antibody), rather than using small actin probes (such as SiR-actin) that directly label F-actin, in a very small cell. We now display the fluorescence images using a different lookup table to increase the visibility and, in addition to the deconvolved Airyscan data, we also optimized our protocol further to allow for super-resolution STED imaging of the immuno-fluorescence experiments. This has been added as Fig. 5h and clearly shows filamentous assemblies, particularly in the longer protrusions.

Other comments/suggestions:

- Line 118: "encoding 5,119 proteins"

R: Changed as suggested.

- Line 121: "and even three ribosomal RNA operons"

R: Changed as suggested.

- Lines 126/136: What are "unique orthologous proteins"? What are they orthologous to?

R: Considering that these genes are unique to our strain, our wording was misleading. We are now using the term orthologous to refer only to proteins that are shared between both *P. syntrophicum* and *L. ossiferum*. Thus, we removed the term when referring to unique proteins.

- Fig. S11: It would be helpful to see the position of eukaryotic actins in this tree.

R: Fig. S11 has been removed and eukaryotic actins as well as bacterial actin-like proteins have all been incorporated into Figure 5d (as requested by Reviewer 3).

Referee #2 (Remarks to the Author):

The manuscript by Rodrigues-Oliveira et al. reports the enrichment, genome, and ultrastructural characterization of *Ca. Lokiarchaeum ossiferum*, the second ever member of Asgard archaea, the closest relatives of Eukaryotes.

This is per se a giant achievement, as we know next to nothing about these intriguing microorganisms.

Moreover, the authors make it very clear how their model offers undeniable advantages over *Prometheoarchaeum* that was isolated in coculture last summer. It thrives in easy to access sites (shallow water sediment in Slovenia), and the authors successfully found optimal growth conditions (7-14 days generation time and high cell densities). But most importantly, the authors take on an incredible technical challenge with imaging the cell ultrastructure of this uncultured archaeon, providing unprecedented insights into its organization.

The quantity and quality of the images is stunning. Also, the authors do not stop there but complement these images with many other techniques, making this work extremely rich and exhaustive, at the limit that can be done on an uncultured prokaryote. I particularly liked the solution to identify *L. ossiferum* with respect to other archaea and bacteria present in the enrichment by linking it to a specific structure of the ribosome, a very nice idea that opens the way to identify Asgard archaea in their natural environment, as FISH cannot be combined with high resolution imaging.

So, overall, I congratulate the authors for this very nice work which we'll be of great impact to the scientific community.

R: We thank the reviewer for the enthusiasm and positive comments.

I have just a few questions/comments:

First, two things that the authors may want to elaborate just a bit further in the text:

- How many archaea have been characterized structurally in such depth and how specific are these protrusions and cell complexity in asgard archaea?

R: While several other archaeal strains have been imaged by cryoET in the past in order to study different cell biological features, the sampling density as compared to bacteria for instance is much lower. The protrusions reported here have certain similarities to complex nanotubes and cell bridges that have been observed e.g. in *Haloferax volcanii*. These studies were/are cited in the main text (ref. 37-40, line 206).

-and with the ribosomal expansion, how do you know that they are not present in yet to identify archaea, and in that case how can you be sure to link if to asgards in the environment?

R: This is a very good point. Our structural approach is based on a combination with metagenomic data and relative abundance information to make sure that none of the other co-isolated species (in an enrichment culture or in a defined environmental sample, e.g. in a microbial mat) possesses similar structural features. To emphasize this, we have added an alignment of the LSU rRNA obtained from the other MAGs of the culture to Extended Data Fig. 6d. Furthermore, we have added the following statement to the text (lines 185-186: "Importantly, expansion segments were not present in the large subunit rRNA sequences of co-cultured species (Extended Data Fig. 6d)."

Line 117: you may want to state here that you obtained a closed genome in one contig, although this is already in the figure it might be good to say it also here.

R: Changed as suggested.

Line 125: the genome is significantly larger than Promethe, which is very close phylogenetically. You may want to reassure further the readers that you did not make any mistake of assembly here. Did you check that they do not have a double of ribosomal proteins for example?

Also, it may be informative to indicate the genome sizes of asgard members on Figure 2e, or even only the lokiarchaea. This can also be briefly stated in the text. Did you sequence the genomes of the other two members (halodesulfo and methanogenium) enrichment other than only the 16S?

R: We are aware of the difference in genome size between both organisms and have taken particular care in checking for contamination or mis-assembly. Although the genome is ~1.5 Mbp larger, the values for completeness and contamination of the *L. ossiferum* genome are similar to that of *Prometheoarchaeum*. When we checked for single copy marker genes, the percentage of potentially duplicated genes for both organisms are in the same range (6% for *Prometheoarchaeum* and 7.9% in *L. ossiferum*). If a large fraction of the genome extension was due to contamination, this should have resulted in a significantly higher value (30% extra according to its larger genome size). Among the single copy marker genes are also the ribosomal proteins. None of them was found to be duplicated.

As suggested, we have now indicated the sizes of the Asgard genomes in Figure 2e and indeed genome sizes vary from 2.6 to 9 Mbp. (However, one has to consider that these genomes are based on MAGs and not closed genomes as is only true for *Prometheoarchaeum* and *L. ossiferum*).

The genomes of the other members in our enrichment (methanogens and *Desulfovibrio*) are currently high-quality MAGs, all at >90 % completeness and low contamination. A future study will focus on their further characterization.

Line 169: refer to figure 3c already at the end of this sentence.

R: Changed as suggested.

Line 173: the authors may want to clarify once more the technical challenges that prevented a correlated FISH-cryoET approach.

R: The FISH protocol involves very harsh treatment (including chemical fixation, SDS and ethanol treatment as well as high temperatures). In our hands, no cellular ultrastructure was preserved (i.e. we could only correlate the signal to cellular debris). To illustrate this better, we have added the following statement to the text: "FISH, however, involves harsh sample preparation steps (chemical fixation, dehydration, permeabilization and high temperatures), which did not allow the preservation of the fragile cellular ultrastructure." (lines 175-177).

Line 266: the Etymology before the conclusions looks a bit bizarre, I would move it further down.

R: Changed as suggested.

Line 281: please nuance a bit the statement that the actin-based cytoskeleton has never been visualized. Or explain better here your advances with respect to the *Prometheoarchaeum* paper.

R: We have revised this statement accordingly (lines 272-273): "We discovered an elaborate actin-based cytoskeleton in Asgard archaea, which has long been hypothesized (e.g. 4–6, 42–44), but never visualized."

Line 298: see my second comment and maybe nuance a bit here or develop further.

We have nuanced the statement and now say: “our study established approaches that will enable imaging of Asgard archaea in a culture-independent manner in environmental samples“ (line 290).

Referee #3 (Remarks to the Author):

Review: Rodrigues-Oliveira et al., "Complex cytoskeleton and cell architecture in an Asgard archaeon", Nature, manuscript: 427877 (2022)

The authors report the enrichment of a new Asgard archaeon, which they name Lokiarchaeum ossiferum. They sequence the complete genome and report a number of interesting statistics and findings derived from the sequence. The authors then progress to imaging and use electron cryo-tomography and some FISH and immunofluorescence.

One difficulty they overcome very elegantly is that the culture contained a number of organisms and they needed to be sure which cells are which in their images. They used "visual proteomics" to achieve this, which, to my knowledge, is the first time the method has been used in this way. They determined the (low resolution) sub-tomogram-averaged structure of the ribosome from the cells they thought were *L. ossiferum* cells and determined that the structure they obtained showed the known expansion segments 9 & 39 in the RNA that are characteristic of Asgard organisms. Most importantly, the resulting tomograms also show extensive and beautiful cellular protrusions on the surface of the cells. Because these are thin, the images are very convincing and it is also possible to discern chemotaxis arrays, ribosomes and filaments in these protrusions and in small cell bodies. They then go on to determine a low-resolution structure of the filaments and produce a map that is consistent with the filaments being F-actin as formed by "lokiactin".

It is important for me to point out two things: first, this work is difficult and only one (low quality) study has previously reported an enrichment culture of an Asgard organism. No previous studies have provided images that show with such clarity and certainty that Asgard organisms can have surface protrusions and that these contain F-actin-like filaments. Second, the protrusions are supportive of a theory of how eukaryotes came about, the now-famous inside-out theory by Baum & Baum (ref 50). It is clear that we have to think about this theory much harder now and it is also clear that experimental progress will now be rapid, not least based on the study under review here.

I review many papers, but I can state that this is the most exciting manuscript I had the pleasure of being asked to review for a number of years: it is important, well-performed, beautiful to look at and easy to follow and understand. The manuscript is well written and presented. The two corresponding authors are experts in their fields (archaeal microbiology and cellular tomography) and the work is of very high quality. While I am very supportive of this getting published asap,

below are a few comments and questions, of which some might seem overly critical, but I wanted to be extra helpful. Please publish.

R: We are delighted to read this positive assessment and thank the reviewer for the constructive comments.

Some comments and questions in no particular order:

1: (title): I am not sure the word "complex" is justified (yet) for the cytoskeleton imaged in the study. A suggestion: "Enriched Asgard archaea reveal cell surface protrusions that contain actin filaments"

R: We have changed the title to: "Actin cytoskeleton and complex cell architecture in an Asgard archaeon".

36: How are the filaments shown here principally different or more “complex” than, let’s say ParM actin-like filaments that cross the body of E. coli cells? (related to previous point)

R: This is a very good point. The filament itself is not more complex (and very similar to Crenactin which we have now added to Fig. 5c to underline this) and the apparent complexity could be a consequence, rather than a cause, of the complex cell architecture. However, the very prominent actin cytoskeleton in combination with the large body of evidence that it is regulated by gelsolins/profilins (as described e.g. in ref. 5,19,24-26) led us to use the term complex.

53: ESCRT-III proteins have just been identified in bacteria ... (PspA, Vipp1 ..) / 54: Intro could mention that bona fide F-actin has been found outside Asgard (crenactin).

R: We now specify that Asgard archaea possess a functional ubiquitin-coupled ESCRT system for membrane remodeling (line 53). We agree that it is important to point out that many “ESPs” have bacterial and archaeal homologs outside of Asgards (even though Asgard archaea appear to possess the closest homologs to the eukaryotic proteins as well as accessory proteins). We therefore also added Crenactin to the introduction (line 56).

65: To my knowledge this model was first mentioned in ref 50 (Baum & Baum) and needs to be mentioned/cited here and not at the very end of the manuscript.

R: We have now mentioned and cited the inside-out model in the introduction (now ref. 28).

65: Why would that theory need ESPs? Could also have been done with proteins whose genes were lost afterwards. Not a good argument.

R: We have revised this statement.” The role of Asgard ESPs could so far not be investigated in the natural host, making it difficult to further test these conceptual models.” (lines 67-69).

93: Those enrichments of 25-80% are impressive – have single cell inoculations been tried after cells were recognised? And also with the putative syntrophic partners?

R: We have not tried single cell inoculations, neither with isolated cells nor together with the syntrophic partners yet. This is an interesting suggestion, that we might follow up on. The current enrichment status is the product of several years of intense cultivation efforts. From our experience, attempts to reach even higher enrichments might take another significant period of time.

108: Unit of 2.5? Unit needs to be mentioned first (minor issue, sorry).

R: Added as suggested.

162: Possible that the fragility of the cells causes them to fragment on the grids? How can we be sure that the cells really look like this when not blotted to near-dryness on EM grids? Superresolution light microscopy with a membrane dye? Cryo slice-and-view (black face imaging)? This is something that could be done quite quickly and I think could add additional certainty.

In order to address this point, we performed imaging by different complementary methods with different sample preparation methods, including SEM, light microscopy of immuno-stained samples and conventional thin section EM. Data from all three imaging modalities indicate consistently the presence of the observed cellular ultrastructure seen in cryoET.

That said, we carefully considered the reviewer’s suggestion. We found that some membrane dyes would stain the cells, however, these stains are not compatible with fixation. Live cell super-resolution imaging of these very small structures (i.e. protrusions of <100nm width) is impossible, since cells cannot be sufficiently immobilized (and imaging has to be done in an

anoxic atmosphere) without causing artifacts. New methodological advances will be needed to observe the biogenesis of the protrusions and address their function. Since currently only fixed samples allow for sufficiently high resolution, we decided to instead show the cellular ultrastructure by SEM.

Cryo-volume imaging (slice-and-view) is unfortunately currently not feasible for multiple reasons: It would involve blotting and plunge freezing just like the cryoET approach. Localizing cells from an unconcentrated sample would not be possible in an SEM and such a sample would also be too thin for cryo slice-and-view experiments (the exposed surface would be too small to image since the front of the milled area shows charging artifacts). We frequently perform these experiments in our lab with bacterial cells and concentrate them to form a layer of multiple cells on the grid. Our attempts to concentrate *L. ossiferum* cells by centrifugation to obtain a thicker “lawn” on the grid were so far unsuccessful because the protrusions of unfixed cells are highly fragile.

To address the reviewer’s comments, we have added more SEM and 2D cryoTEM images to Extended Data Fig. 7 d/e, which further underline the variability of the cell shape and emphasize the consistency of datasets from different imaging modalities.

173: What were those challenges that prevented FISH-CLEM? Would have been nice and powerful ...

R: Please see also our response to Reviewer 2. We agree that FISH-cryoET would be a very powerful approach, but current FISH protocols are too harsh, so that the ultrastructure of fragile cells (but in our experience also many bacterial cells) cannot be preserved. In our experiments, we could only find cell debris after FISH, which did not allow for ultrastructural characterization or even identification. We have included a statement in the main text (lines 175-177).

177: How many particles were averaged from how many cells? (Methods say 4,126 particles, and this should be mentioned in the main text).

R: We have added this information to the main text (lines 180-181).

Also, it seems from the methods that no ribosomes were reconstructed from the non-*L. ossiferum* cells? Is that not a (theoretical) weakness (lack of control)?

R: This is a very good point. We did not reconstruct an average from the other archaeal cells (they were also a lot less abundant) but instead relied on metagenomic information and compared the LSU rRNA of the most abundant organisms to show that the expansion segments are indeed unique to *L. ossiferum* in the enrichment culture. The technique therefore relies on a combination of metagenomics and visual proteomics. We have included the alignment to Extended Data Fig. 6 and added a statement to the main text (lines 185-186): “Importantly, expansion segments were not present in the large subunit rRNA sequences of co-cultured species (Extended Data Fig. 6d).”

191: I am not sure what the scanning EM adds here. Sample preparation is harsh. Volume EM (cryo slice-and-view/blockface imaging) would have been better ...

R: Please see our comment above. Cryo slice-and-view would unfortunately induce more artifacts due to the necessity to concentrate the unfixed sample.

204: Filament branching is exciting! Any signs of branching proteins in the genome

R: We have analyzed the genome for the presence of the seven subunits of the ARP2/3 complex that is involved in actin branching in eukaryotes. We did not find any homologs, except for the two proteins ARP2 and ARP3, of which Lokiactin is a homolog (see Figure 2). The very few examples of putative branch sites could of course also be overlapping filament ends and

we did not observe a characteristic branching angle (as for eukaryotic ARP2/3). To avoid confusion, we have removed the statement.

212: Surface structures: where *L. ossiferum* cells contact other cells, any sign that these structures are involved?

R: This would be very exciting, currently we only see these contacts in large “clumps” (as shown in Extended Data Fig. 7c), which are quite thick. The thick ice, also in the periphery, did not yet allow us to obtain interpretable tomograms of cell-cell interactions.

227: I think it would only be justified to suggest that the filaments scaffold the protrusions if filaments were found in most, if not all membrane structures. More careful wording I think is needed here. It is quite possible that the protrusions are caused by some other proteins and the filaments just happen to be inside to do something else, for example.

R: We agree that there is no evidence for Lokiactin filaments shaping the membrane yet. Scaffolding is in our view a very reasonable explanation (e.g. to prevent the collapse of the very thin membrane tubes). The protrusion biogenesis would not have to rely on Lokiactin, which could subsequently act as a scaffold. We have removed “re-shaping” (line 235) and only talk about scaffolding since this is our most likely explanation for the observation. We are trying to perform functional experiments, but these will be a lot more difficult to interpret in such a challenging experimental system.

251: A much more involved way to make sure the raised antibodies do what they are supposed to do would have been to make the various actins recombinantly, and to see how they bind. Not suggesting that this needs to be done, just a suggestion.

We attempted the expression of the actin homologs in *E. coli*, however, we were unable to generate full-length protein for all homologs. A preliminary report from others in a review (<https://pubmed.ncbi.nlm.nih.gov/33049465/>) showed that expression of Lokiactin required the use of insect cells. In the time frame of the revisions, we were not able to generate heterologously expressed protein.

That said, we now show the high level of transcription of Lokiactin (Fig. 5e), as well as STED imaging of the immuno-fluorescence experiments (Fig. 5h), which both strengthen our conclusions.

254: And FISH-CLEM again was not possible? Why?

R: IF/cryoCLEM might not be very conclusive, since cells have to be fixed and permeabilized. We have tried to perform CLEM with live-cell actin stains but were unable to locate cells, most likely due to cytotoxic effects of the stains. Finding untreated *L. ossiferum* cells on a grid currently requires overnight screening at low magnification (for just a few targets), so even a small drop in cell viability makes imaging impossible.

258-264: Not sure this is needed. It is speculative and the existence of gelsolin and profilin homologues could just be mentioned here.

R: We have shortened this paragraph accordingly.

266: *L. ossiferum*: don't the authors suggest that most likely all Asgard organisms have actin filaments? Does the new name make sense in light of this since it will probably not be a distinguishing feature?

R: Often microbes are named based on striking characteristics that are described for the first time but that are not necessarily distinguishing from all other organisms. For example, as in

Sulfolobus acidocaldarius, the type species of *Sulfolobales*, a genus comprising almost exclusively acidophilic thermophiles.

280: I object to the assertion that the first actin-cytoskeleton has been discovered, if that was meant. Of course, actins are present throughout the bacteria and also archaea, and crenactin forms F-actin filaments that are structurally indistinguishable from eukaryotic F-actin. Please re-phrase (I know the second sentence is meant to be conditioned by the first, but this is not made very clear I think).

R: The main text was revised accordingly (line 272-273): “We discovered an elaborate actin-based cytoskeleton in Asgard archaea, which has long been hypothesized, but never visualized.”

291-293: Again, I am not sure I would necessarily take away from the pictures that *L. ossiferum* actin shapes membranes and is a scaffold (for example Figure 4j). More careful wording is probably safer right now.

R: We agree that Lokiactin is probably not solely responsible for membrane shaping, since we observe protrusions without filaments. “Scaffolding” does not imply an active membrane shaping function. Our tomograms and also the new IF/STED data shows extensive filaments that traverse very long-range protrusions, consistent with the idea of a scaffold that maintains the cell shape. We have removed the membrane shaping statement from the main text but still believe that scaffolding is a very likely explanation (also because it is most likely necessary to stabilize the very fragile protrusion networks).

304: The authors mention earlier that they think it is unlikely that cell-cell contacts are needed for syntrophy. That weakens the statement here, I think. And, why are no cell contacts shown by cryo-ET? Too few?

R: We unfortunately only observe cell-cell contacts in areas that are too thick to image. We cannot exclude that cells rely on cell-cell contacts in the actual culture which are then disrupted during sample preparation, or whether these contacts are required during certain lifecycle stages (as stated in line 194). It is likely that contacts are a lot more permanent in the sediment, which we currently cannot study.

307: The Baum a & Baum model (ref 50) could be described in more detail since the data shown here is very supportive of what that theory predicted (even the picture!).

R: We agree with the suggestion and have revised the main text to explicitly point out this remarkable similarity (line 300-302): “These findings strongly support a gradual path of mitochondrial acquisition through protrusion-mediated cell-cell interactions, which have been proposed previously in the “inside-out” and “E3” hypotheses.”

310: How would compartmentalisation work? The protrusions would need to be sealed off, no? Any evidence for that?

R: Compartmentalization of cellular processes can be mediated by very different mechanisms that do not always rely on a membrane-enclosed compartment. Cellular “protrusions” as e.g. eukaryotic cilia or neuronal axons lead to the compartmentalization of cellular processes mediated by active transport. Whether or not similar mechanisms (e.g. active transport, localized translation etc.) exist in *L. ossiferum* remains an interesting open question. The protrusions observed here did not reveal evidence for a “seal”.

355: Data availability: will raw images that were used to reconstruct tomograms be deposited in EMPIAR? This would be really helpful, thank you!

R: Representative tomograms and tilt series will be uploaded to EMDB and EMPIAR.

619-: Wouldn't it have been better to do the structure of the filaments in Relion 4 directly from the pseudo sub-tomograms (no helical symmetry or only later). The projection analysis used will severely degrade resolution as all the errors that went in during initial tomogram reconstruction will persist, if I understood the method correctly.

R: Thank you for the suggestion. We previously could not obtain a reliable sub-tomogram average due to the challenging properties of the sample (very low SNR and very limited dataset of good particles). We now however improved our workflow by using the helical parameters obtained before for the 3D refinement in Relion 4. We were able to obtain an average with slightly improved resolution and refined helical parameters. We have summarized this approach in Extended Data Figure 9d and the Methods. We could not further improve this without overfitting and would likely need significant advances in sample preparation to obtain better results.

Figure 5d: Since crenactin has the same structure as F-actin, I think it is slightly unhelpful to make it an outgroup. F-actin exists outside Asgard. Bacterial actins and other actins may need to be included to not give the wrong impression. Figure S11 attempts this, but again it only looks at sequences and disregards structural conservation, which is more important for function and evolution, since sequences diverge more quickly than structures.

R: A new tree has been calculated by including homologs from the crenactin, MreB, MamK and ParM groups (Fig. 5d).

General question: would you dare speculate why the protrusions are there? What is their function? They are not there to make eukaryotes, surely ...

R: We currently do not have solid data on this topic. However, we do observe very extensive protrusions when cells occur in multi-species clumps, so they could very well be involved in cell-cell interactions, especially in biofilms that can probably be found in the sediment. We hope to optimize our sample preparation protocols to soon be able to image these interactions.

Figure S10: no protrusions visible here? And, as is often the case, the immunogold labelling does not add much certainty ...

R: We now included a quantification of immunogold labeling experiments (Supplementary Table 10). While protrusions can be seen in all images of Extended Data Fig. 10, the slices do not always reveal the connection of the protrusion to the cell body (thin section TEM).

Reviewer Reports on the First Revision:

Referees' comments:

Referee #1 (Remarks to the Author):

In the revised version of the manuscript, the authors have fully addressed my concerns. This is now a beautiful and convincing paper that for the first time provides detailed insight in the actin cytoskeleton of Asgard archaea and thus illuminates the origin of the complex actin cytoskeleton found in eukaryotic cells today. I am sure it will be received with great interest by the scientific community and beyond.

Referee #2 (Remarks to the Author):

The reviewers have answered all my comments and I am satisfied with the revised manuscript. The article is suitable for publication.

Referee #3 (Remarks to the Author):

Re-review: Rodrigues-Oliveira et al., "Actin cytoskeleton and complex cell architecture in an Asgard archaeon", Nature, manuscript: 427877r1 (2022)

The manuscript has been significantly improved and as far as I can see all comments from me and the other two reviewers have been considered carefully. In particular:

- 1) The title change is good.
- 2) My main concern about the lokiactin 3D reconstruction has been sufficiently alleviated. The new reconstruction is cleaner and allows the 4 subdomains structure of the subunits to be seen.
- 3) The addition of lokiactin transcription data is very helpful.
- 4) The comments about why FISH-CLEM, volume EM and the making of recombinant lokiactins did not work are convincing.
- 5) More TEM views convince me sufficiently that the protrusions shown are frequent and a bona fide property of the cells observed.
- 6) The issue of not mentioning crenactin enough has been resolved well.
- 7) Biotium makes covalent membrane stains that can be fixed, if the authors would like to investigate this further.
- 8) Many minor points have been dealt with well, mostly dealing with imprecise language.

In summary, I remain of the opinion that this is important, timely and exciting work that should be published asap.